# Astrocytes distress triggers brain pathology through induction of δ secretase in a murine model of Alzheimer's disease

Vanessa Schmidt [1] ✉, Ewelina Ziemlinska [2], Tomasz Obrebski [2], Jemila P. Gomes[3], Ewa Zurawska-Plaksej[4], Jaroslaw Cendrowski [5,6], Johan Palmfeldt [7], Barbara L. Hempstead[8], Thomas E. Willnow [1,3] ✉ & Anna R. Malik [2] ✉

The importance of astrocytes for Alzheimer's disease (AD) pathology is increasingly appreciated, yet the mechanisms whereby this cell type impacts neurodegenerative processes remain elusive. Here we show that, in a genetic mouse model with diminished astrocyte stress response, even low levels of amyloid-β trigger astrocyte reactivity, resulting in brain inflammation and massive amyloid and tau pathologies. This dysfunctional response of astrocytes to amyloid-β acts through activation of δ secretase, a stress-induced protease implicated in both amyloid and tau-related proteolytic processing. Our findings identify a failed astrocyte stress response to amyloid-β as an early inducer of amyloid and tau co-morbidity, a noxious process in AD acting through a non-canonical secretase pathway.

Recent findings have identified the significance of glia for healthy brain aging, and disturbances in non-neuronal cell types as drivers of neurodegenerative processes in Alzheimer's disease (AD). The underlying concepts have best been elucidated for microglia that play essential roles in phagocytic clearance of amyloid deposits as well as inflammatory processes in the AD brain (reviewed in refs. 1,2). By contrast, the relevance of astrocytes, the major non-neuronal cell type in the brain for AD remains less well understood. Possible contributions to AD pathology are supported by recent evidence that astrocytes are sensitive to exposure to amyloid-β peptides (Aβ)[3,4] and that astrocyte stress causes neuronal dysfunction and death[5–7]. Also, the pro-inflammatory actions of reactive astrocytes in the AD brain likely contribute to progression of AD pathologies[8–11].

To elucidate molecular concepts of astrocytes (dys)function in AD, we focused on the sortilin-related receptor CNS expressed (SORCS) 2, a member of the VPS10P domain receptor gene family of intracellular sorting receptors[12]. In neurons, SORCS2 is best known for

its role in control of neuronal cell fate decision and plasticity, functions linked to the ability of the receptor to control neurotrophin-dependent neurite outgrowth, growth cone collapse, and dopaminergic wiring[13–16]. With relevance to this study, SORCS2 has also been genetically linked to sporadic AD, an association possibly explained by increased amyloidogenic processing seen when inactivating *SORCS2* in HEK293 cells[17].

An alternative concept to potentially explain the relevance of SORCS2 for brain health and disease relates to recent work, documenting SORCS2 action in protective stress responses in multiple mammalian tissues. In pancreatic α-cells, SORCS2 directs secretion of osteopontin, a factor that stabilizes insulin release from β-cells under glucose stress[18]. In neurons, SORCS2 sorts the amino acid transporter EAAT3 to the cell surface, increasing cysteine import and glutathione production as means of oxidative stress response in epilepsy[19]. Finally, in astrocytes, the receptor facilitates release of endostatin to promote angiogenesis in the post-stroke brain[20]. Assuming a similarly protective

[1]Max-Delbrueck-Center for Molecular Medicine in the Helmholtz Association, Berlin, Germany. [2]Faculty of Biology, University of Warsaw, Warsaw, Poland. [3]Department of Biomedicine, Aarhus University, Aarhus, Denmark. [4]Department of Toxicology, Wroclaw Medical University, Wroclaw, Poland. [5]Laboratory of Cell Biology, International Institute of Molecular and Cell Biology, Warsaw, Poland. [6]Department of Experimental Oncology, Maria Sklodowska-Curie National Research Institute of Oncology, Warsaw, Poland. [7]Department of Clinical Medicine, Aarhus University, Aarhus, Denmark. [8]Weill Cornell Medical College, New York, NY, USA. ✉e-mail: Vanessa.Schmidt-Krueger@mdc-berlin.de; tew@biomed.au.dk; ar.malik@uw.edu.pl

function for SORCS2 in astrocyte stress response to Aβ, we investigated mouse models of AD lacking receptor expression. We hypothesized that lack of SORCS2 may sensitize astrocytes to amyloid burden and accentuate AD pathologies linked to astrocyte distress in the aging brain. In line with our hypothesis, SORCS2 deficiency aggravated Aβ-induced stress and astrocyte cell death in vitro and in vivo, causing massive amyloid and tau pathologies in murine AD models. Astrocyte dysfunction was linked to aberrant activation of δ secretase, a stress-induced protease implicated in both amyloidogenic and tau-related proteolytic processing. Our findings identify astrocyte stress from Aβ as a potential mechanism of amyloid and tau comorbidities, noxious processes acting through a non-canonical secretase pathway.

## Results

### SORCS2 deficiency sensitizes astrocytes to amyloid-induced cell death

Initially, we tested our assumption that deficiency for the stress response factor SORCS2 may impact the ability of astrocytes to cope with noxious stimuli from Aβ peptides. Treatment of primary murine astrocytes with Aβ-conditioned medium from SY5Y cells, over-expressing human APP[21], induced *Sorcs2* transcript levels, linking receptor expression with Aβ sensing (Fig. 1a). When exposed to Aβ-conditioned medium, primary astrocytes from mice carrying a targeted disruption of *Sorcs2* (KO)[16] showed significantly increased clearance of $A\beta_{40}$ and $A\beta_{42}$ compared to SORCS2-expressing wildtype (WT) cells (Fig. 1b). Internalization of Aβ resulted in intracellular accumulation of the peptide (Fig. 1c), which was more pronounced in mutant than wildtype astrocytes (Fig. 1d). Intracellular accumulation of Aβ coincided with elevated levels of glial fibrillary acid protein (GFAP) (Fig. 1e) and a decrease in lysosomal acidification (Fig. 1f), stress responses more pronounced in KO than WT astrocytes. Ultimately, primary KO astrocytes reacted to prolonged Aβ exposure with decreased viability as shown by PrestoBlue assay (Fig. 1g). Reduced viability coincided with enhanced apoptosis as judged from increased levels of cleaved forms of poly ADP-ribose polymerase (cl. PARP) and cleaved caspase-3 in KO as compared to WT astrocytes (Fig. 1h–i). In brain autopsies from AD patients, levels of *SORCS2* transcript were significantly increased when compared to control subjects, correlating amyloid burden with enhanced receptor expression in humans as well (Fig. 1j). Taken together, these data supported a presumed role for SORCS2 in mitigating noxious insults imposed on astrocytes by Aβ, and they suggested SORCS2 deficiency as a useful experimental paradigm to investigate consequences of heightened astrocyte distress in AD.

### Loss of SORCS2 results in amyloid and tau co-morbidity in a mouse model of AD

To interrogate the impact of enhanced astrocyte stress on AD pathology, we crossed $Sorcs2^{-/-}$ mice with transgenic animals expressing the human $APP^{Ind}$ transgene under control of the platelet-derived growth factor β promoter (PDAPP strain), an established model of AD[22].

The resulting (PDAPP x $Sorcs2^{-/-}$) mice are referred to as PDAPP/KO herein. They were compared to sex- and age-matched (PDAPP x $Sorcs2^{+/+}$) animals (PDAPP/WT).

PDAPP/KO females showed an age-dependent increase in brain cortex levels of soluble $A\beta_{40}$ and $A\beta_{42}$ compared to PDAPP/WT mice, starting from 25 weeks of age (Fig. 2a, b). At 20 weeks of age, cortical $A\beta_{40}$ and $A\beta_{42}$ levels displayed a bimodal distribution in the PDAPP/KO cohort with half of the animals having a 2-fold but the other half an approximately 9-fold increase over PDAPP/WT (Fig. 2c). At 40 weeks of age, all PDAPP/KO mice showed massive accumulation of $A\beta_{40}$ and $A\beta_{42}$ with cortex levels exceeding those in PDAPP/WT by more than 10-fold for $A\beta_{40}$ and 30-fold for $A\beta_{42}$ (Fig. 2d). The same bimodal

distribution at 20 weeks, and a massive accumulation in all mutant mice at 40 weeks of age, was observed for soluble $A\beta_{40}$ and $A\beta_{42}$ in hippocampal extracts (Fig. 2e, f). Accumulation of Aβ resulted in a pronounced age-dependent increase in senile plaque deposition in cortex and hippocampus of aging PDAPP/KO females, as documented by thioflavin S staining (Fig. 2g, h) or Aβ immunodetection (Fig. 2i, j).

Remarkably, loss of SORCS2 also caused tauopathy-like phenotypes in aging PDAPP/KO females. In detail, SORCS2 deficiency increased levels of murine tau, phosphorylated at inorganic pyrophosphatase 2 (PPA2) target sites $Ser_{202}$ and $Thr_{205}$ (antibody AT8), in cortex (Fig. 3a) and hippocampus (Fig. 3b) of aged PDAPP/KO mice, as shown by Western blotting. Levels of murine $ptau_{Thr231}$ (AT180 antibody) were also increased, while the levels of total tau (HT7 antibody) were unchanged compared to control females (Fig. 3a, b). A relative increase in the ratio of $ptau_{Thr231}$ over total tau in cortex and hippocampus of 40 weeks old PDAPP/KO females was confirmed by ELISA (Fig. 3c, d). Both $ptau_{Ser202/Thr205}$ and $ptau_{Thr231}$ variants contribute to formation of tau tangles[23–27]. Immunohistology confirmed elevated levels of $ptau_{Ser202/Thr205}$ immunoreactivity in several brain regions of 40 weeks old PDAPP/KO compared to PDAPP/WT females (Fig. 3e, f). Immunoreactivity for total tau (HT7) or cleaved $tau_{Asp421/Asp422}$, which rapidly assembles into filaments[28,29], showed a distinct fibrillary appearance in 40 weeks old PDAPP/KO brains, not seen in PDAPP/WT tissue (Fig. 3g, h). Co-immunostaining of total tau with cell type-specific markers documented more pronounced colocalization with endothelial cell marker CD31, than with markers of neurons (MAP2), astrocytes (GFAP), or microglia (IBA1) (Fig. 3i). Association of tau aggregates with the brain vasculature has also been reported in other murine models and patients with tauopathies[30,31].

Importantly, levels of $ptau_{Ser202/Thr205}$ and $ptau_{Thr231}$ were unchanged in cortex (Fig. 4a, b) and hippocampus (Fig. 4c, d) of aged female KO mice lacking the PDAPP transgene. Also, the ratio of $ptau_{Thr231}$ over total tau in cortex and hippocampus was comparable to that in WT tissues (Fig. 4e, f). These findings documented that stress from human Aβ in the PDAPP line was necessary to trigger tau hyperphosphorylation and aggregation in SORCS2-deficient females. Massive accumulation of $A\beta_{40}$ and $A\beta_{42}$ in cortex and hippocampus was also seen in PDAPP/KO males at 40 weeks of age (Supplementary Fig. 1a). Contrary to females, no significant increases in $ptau_{Ser202/Thr205}$ and $ptau_{Thr231}$ levels (Supplementary Fig. 1b, c) nor change in $ptau_{Thr231}$/tau ratio (Supplementary Fig. 1d, e) were seen, in line with the sexual dimorphism observed for phosphorylation of brain tau[32,33].

### Aβ stress induces reactivity and loss of astrocytes in SORCS2-deficient mice

Next, we investigated the cellular consequences of amyloid and tau co-morbidity in our AD mouse model. Using a cell death assay, no difference in viability was seen comparing cortical or hippocampal extracts of WT and KO female brains lacking $APP^{Ind}$ (Fig. 5a). Cell death increased in both genotypes in the presence of the PDAPP transgene, but this increase was significantly higher in PDAPP/KO than PDAPP/WT females at 40 weeks of age (Fig. 5a), indicating heightened sensitivity of aging SORCS2-deficient brains to Aβ-induced cell death.

To identify the cell type(s) most impacted by Aβ stress, we compared the cell type composition of brains in aged WT and KO females with or without PDAPP transgene. To do so, we established a FACS protocol to isolate pure populations of individual cell types, including neurons, astrocytes, microglia, and oligodendrocytes (Supplementary Fig. 2a, b, c). RT-PCR on sorted cell populations from WT animals lacking PDAPP documented *Sorcs2* transcripts to localize to astrocytes, neurons, and endothelial cells, both in cortex and hippocampus (Fig. 5b). Remarkably, *Sorcs2* transcript levels significantly increased in astrocytes, but not neurons or endothelial cells, in the presence of the PDAPP transgene (PDAPP/WT, Fig. 5b), corroborating induction of

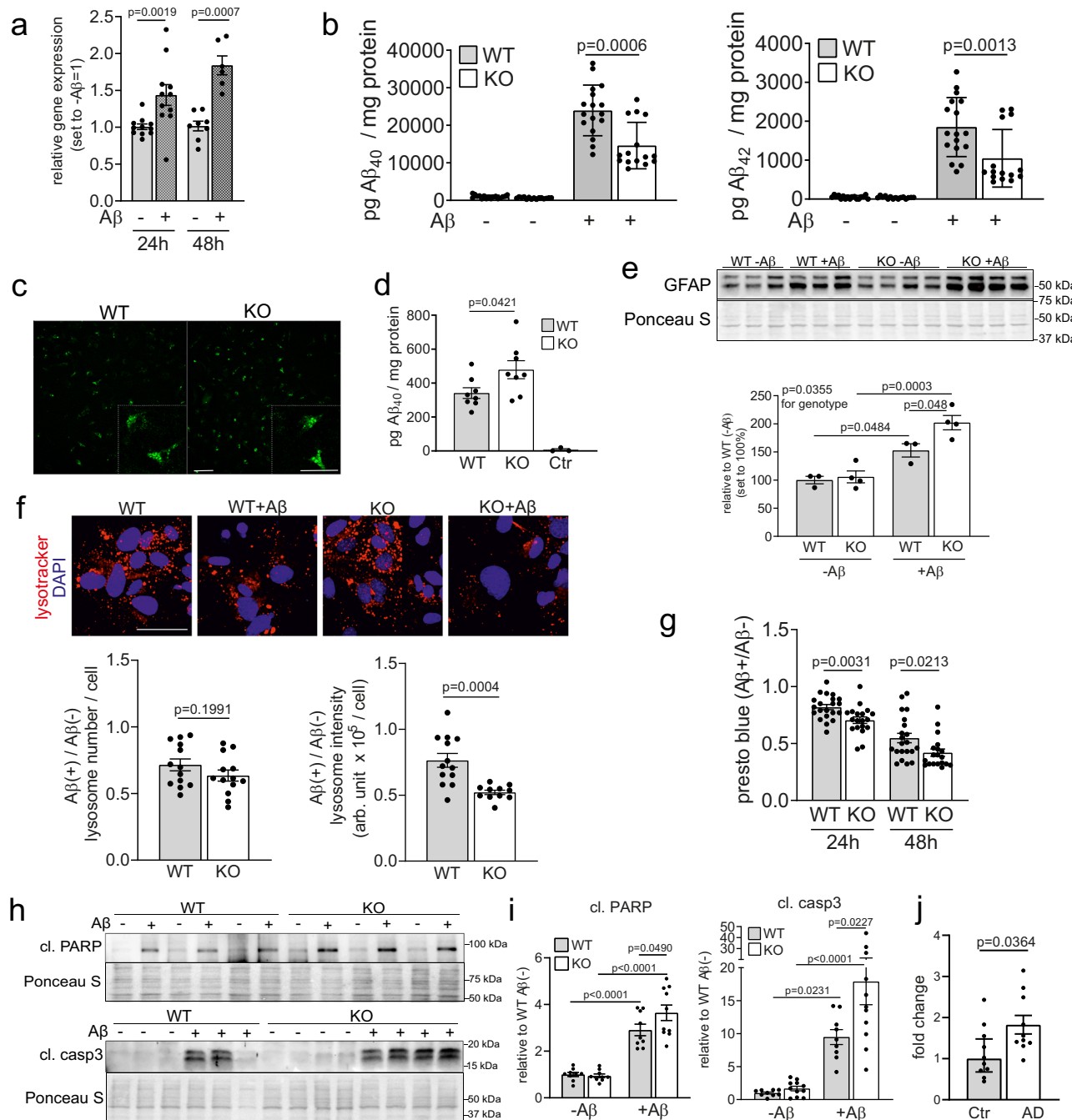

receptor expression as a unique response of astrocytes to amyloid stress. This transcriptional response was an early feature of amyloid stress, as concluded from comparable *Sorcs2* transcript levels in PDAPP/WT animals at 20 and 40 weeks of age (Supplementary Fig. 3a, b). SORCS2 expression in astrocytes and neurons in cortex and hippocampus was also confirmed by co-immunostaining with cell type specific markers S100β and NeuN (Fig. 6).

Comparative cell counts in PDAPP/KO and PDAPP/WT females at 40 weeks of age documented a relative loss of astrocytes and endothelial cells, and a concomitant increase in microglia and macrophages in SORCS2-deficient brains (Fig. 5c). A relative increase in microglia and macrophages was already evident in PDAPP/KO females at 20 weeks of age (Fig. 5d), indicating expansion of inflammatory cell types to precede fulminate amyloid and tau pathologies, as well as associated cell death, seen at 40 weeks of age. No changes in relative numbers of microglia and macrophages, or other sorted cell types,

were observed in cortex or hippocampus of 40 weeks old female KO lacking PDAPP when compared to WT controls (Fig. 5e, f).

As SORCS2 deficiency increased the sensitivity of primary murine astrocytes to Aβ stress in vitro (see Fig. 1), we focused further analyses on this cell type in PDAPP/KO mice in vivo. Immunostaining for astrocyte cell surface antigen-2 (ACSA2) confirmed an overall reduction in astrocyte immunoreactivity in PDAPP/KO female brains at 40 weeks of age (Fig. 5g). To interrogate the astrocyte subpopulation lost, cells were analyzed by flow cytometry using a combination of markers GFAP, aldehyde dehydrogenase 1 family member L1 (ALDH1L1), S100β, ACSA2, and SOX9 (Supplementary Fig. 3c). Reduced cell numbers were observed for all astrocyte subpopulations with exception of the S100β+/ACSA2- fraction, which includes astrocytes but also oligodendrocytes (Supplementary Fig. 3d). While aged PDAPP/KO brains contained fewer GFAP+ cells, GFAP levels in the remaining astrocytes were increased compared to PDAPP/WT brains,

**Fig. 1 | SORCS2-deficient primary murine astrocytes exhibit enhanced cell stress in response to Aβ. a** Levels of *Sorcs2* transcripts as determined by qRT-PCR in primary astrocytes from wildtype (WT) mice treated with control (-) or Aβ-conditioned media (+) for 24 h or 48 h. Aβ-conditioned medium had been generated from SY5Y cells stably overexpressing human APP[21]. Data are expressed as relative expression levels (set to 1 for Aβ-) and given as mean ± SEM. *n* = 6 (48 h Aβ +), *n* = 8 (48 h Aβ−), *n* = 11 (24 h) biological replicates per group. Data were analyzed with two-sided unpaired Mann-Whitney U test. **b** Primary WT or SORCS2-deficient (KO) astrocytes were treated with control (-) or Aβ-conditioned medium (+) and residual levels of Aβ were measured after 24 h using ELISA. Data are given as mean ± SEM. *n* = 17 (WT), *n* = 15 (KO) biological replicates. Data were analyzed with two-sided unpaired Mann-Whitney U test. **c, d** Primary WT and KO astrocytes were treated with 1 μM HiLyte-Aβ40 for 6 h. Aβ uptake was visualized by fluorescence microscopy for HiLyte-Aβ40 (**c**, green) and quantified by ELISA in cell lysates (**d**). In (**d**), intracellular Aβ levels are higher in KO as compared to WT cells. No Aβ is detected in WT cells treated with control medium (Ctr, *n* = 3 biological replicates). Data are given as mean ± SEM from *n* = 8 animals per group (two-sided unpaired Student's *t*-test). Scale bar: 50 μm (inset: 200 μm). **e** Western blot analysis of glial fibrillary acidic protein (GFAP) levels in WT and KO astrocytes after 48 h of treatment with control (-) or Aβ-conditioned (+) medium (upper panel) and quantification thereof (lower panel). Staining of the membrane with Ponceau S served as control for equal loading. Data are expressed as levels relative to WT Aβ− (set to 100%) and given as mean ± SEM. *n* = 3 (WT), *n* = 4 (KO) biological replicates per condition (ordinary two-way ANOVA with Tukey's multiple comparisons test to test significance for genotype). **f** Primary WT or KO astrocytes were cultured for 24 h in the presence or absence of Aβ-conditioned medium followed by live staining with LysoTracker Red dye (lysotracker). Lysotracker fluorescence signals (upper panel, red) were used to quantify lysosome numbers as well as cumulative signal intensity per cell (lower panel; in arbitrary units, arb.units). Data are given as mean ± SEM from *n* = 13 (lysosome number; lysosome intensity, WT), *n* = 11 (lysosome intensity, KO) biological replicates per genotype (two-sided unpaired Student's *t*-test). Scale bar: 50 μm. **g** Cell viability was measured using PrestoBlue in primary astrocytes treated with control or Aβ-containing medium for 24 h and 48 h. Data are given as mean ± SEM of fluorescent signal ratio (+Aβ)/(-Aβ) from *n* = 21 (WT) and *n* = 19 (KO) biological replicates (two-sided unpaired Student's *t*-test). **h, i** Western blot analyses of cleaved poly (ADP-Ribose) polymerase 1 (cl. PARP) and cleaved caspase-3 (cl. casp3) levels in WT and KO astrocytes after 24 h (cl. PARP) or 48 h (cl. casp3) of treatment with control (-Aβ) or Aβ medium (+Aβ). Exemplary blots from 3 independent experiments per condition (**h**) as well as quantification from densitometric scanning of replicate blots (**i**) are given. Staining of the membranes with Ponceau S served as control for equal loading. Levels are expressed as relative to WT -Aβ (set to 1), and are given as mean ± SEM from *n* = 9 (cl. PARP, WT; cl. PARP, KO-Aβ; cl. casp3, WT), *n* = 10 (cl. PARP, KO +Aβ), *n* = 11 (cl. casp3, KO) biological replicates (ordinary Two-way ANOVA with Holm-Sidak´s multiple comparisons test). **j** Levels of *SORCS2* transcripts as determined by qRT-PCR in cortex autopsies from AD patients and control subjects (Ctr). Data are expressed as relative expression levels (Ctr set to 1) and given as mean ± SEM from *n* = 10 individuals per group (two-sided unpaired Student's *t*-test). This experiment was performed once. *p*-values for all statistically significant differences are indicated on the graphs. Source data are provided in the Source Data file.

as shown by flow cytometry (Fig. 5h). An increase in GFAP levels in the remaining astrocytes was also documented by Western blot (Fig. 5i) and immunohistology (Fig. 5j, k), arguing for astrogliosis in aged PDAPP/KO brains. Astrogliosis was not seen in 40 weeks old KO females lacking PDAPP, as judged from brain GFAP levels being comparable to that in matched WT (Supplementary Fig. 3e).

## Aβ induces microglia activation and pro-inflammatory responses in SORCS2-deficient mice

Activation of microglia and macrophages represents an important aspect of the inflammatory brain response to amyloid deposits (reviewed in refs. 34,35). Our FACS analyses documented an overall increase in inflammatory cell types in aging PDAPP/KO female brains with a relative shift from microglia to macrophages seen in cortex and hippocampus (Fig. 7a). This phenotype was already observable in 20 weeks old PDAPP/KO females (Fig. 7b), but not in 40 weeks old KO females lacking PDAPP (Fig. 7c). These data substantiated Aβ-induced changes in inflammatory cell type composition as an early feature of SORCS2-deficient brains. Resident microglia and macrophages in PDAPP/KO brains were characterized by an increase in IBA1 expression as shown by immunohistology (Fig. 7d, e) and Western blot (Fig. 7f, g). A shift towards a pro-inflammatory profile was supported by reduced transcript levels for *Cd163* and *Cd206*, characteristic of anti-inflammatory microglia (Fig. 7h, i)[36,37]. Induction of an inflammatory brain milieu in PDAPP/KO mice was further corroborated by cytokine profiling, documenting an increase in pro-inflammatory cytokines and chemokines (Supplementary Table 1). These changes were most obvious in the cohort of young PDAPP/KO mice with low Aβ levels, and seen to a lesser extent in young or aged PDAPP/KO animals with high brain Aβ load. These findings further argued that a pro-inflammatory milieu in PDAPP/KO brains was an early feature of SORCS2 deficiency, preceding full-blown amyloid and tau co-morbidity observed at a later stage.

## Astrocyte-specific loss of SORCS2 recapitulates some features of global receptor deficiency

To dissect cell-type specific actions of SORCS2 in the context of Aβ stress, we generated mouse models with conditional inactivation of *Sorcs2* in either neurons or astrocytes. In detail, we crossed mice homozygous for the floxed *Sorcs2* allele (loxWT)[15] with transgenic lines constitutively expressing Cre recombinase under control of the pan-neuronal *Baf53b* promoter[38] or carrying a tamoxifen-inducible Cre-ERT2 transgene driven by the astrocyte-specific *Aldh1l1* promoter[39]. Both (loxWT x Cre) lines were bred with the PDAPP strain, referred to as neuron-specific (PDAPP/nsKO) or astrocyte-specific (PDAPP/asKO) AD lines herein. Western blot analyses documented a significant reduction in SORCS2 levels in cortical and hippocampal extracts from PDAPP/asKO as compared to PDAPP/loxWT female mice at 38 weeks of age (Supplementary Fig. 4a). Using qRT-PCR on sorted brain cells, loss of expression was attributed to reduced *Sorcs2* transcript levels in cortical and hippocampal astrocytes, but not in neurons (Supplementary Fig. 4b). Astrocyte-specific loss of *Sorcs2* transcript and protein levels could also be induced in younger mice (32 weeks of age, Supplementary Fig. 4c, d). In 32 weeks old PDAPP/asKO mice, soluble Aβ levels initially decreased in the hippocampus when compared to matched PDAPP/loxWT (Fig. 8a), mirroring phenotypes of increased Aβ clearance in primary astrocytes derived from newborn KO mice (see Fig. 1b). This trend reversed in aged PDAPP/asKO animals (38 weeks of age) when hippocampal levels of Aβ40 and Aβ42 increased compared to age-matched PDAPP/loxWT (Fig. 8b), recapitulating the impact of global SORCS2 deficiency in aging PDAPP/KO mice. These distinct effects of astrocytic SORCS2 deficiency on Aβ levels in the hippocampus were less pronounced in the cortices of PDAPP/asKO females at 38 weeks of age (Fig. 8a, b), possibly reflecting differences in astrocyte subtype composition[40] or vulnerability[41] seen in these two distinct brain regions.

Loss of SORCS2 expression was also seen in cortices of PDAPP/nsKO females (Supplementary Fig. 4e). However, *Sorcs2* transcript levels were reduced in both astrocytes and neurons in these animals (Supplementary Fig. 4f). By contrast, neuron-specific *Sorcs2* inactivation was achieved in males as documented by a significant reduction in cortical protein levels (Supplementary Fig. 4e) that resulted from a decrease in *Sorcs2* transcripts in sorted neurons but not astrocytes (Supplementary Fig. 4 f). Still, neuron-specific loss of SORCS2 did not impact cortical or hippocampal levels of Aβ peptides in aging PDAPP/nsKO males compared to matched PDAPP/loxWT animals (Fig. 8c).

Astrocyte-specific *Sorcs2* inactivation did not impact levels of total tau, ptauAT8, ptauAT180, or ptau/tau ratios in cortex or hippocampus, as shown by Western blotting (Supplementary Fig. 5a, b) and ELISA (Supplementary Fig. 5c, d). However, recapitulating additional features of global SORCS2 deficiency, astrocyte-specific loss of SORCS

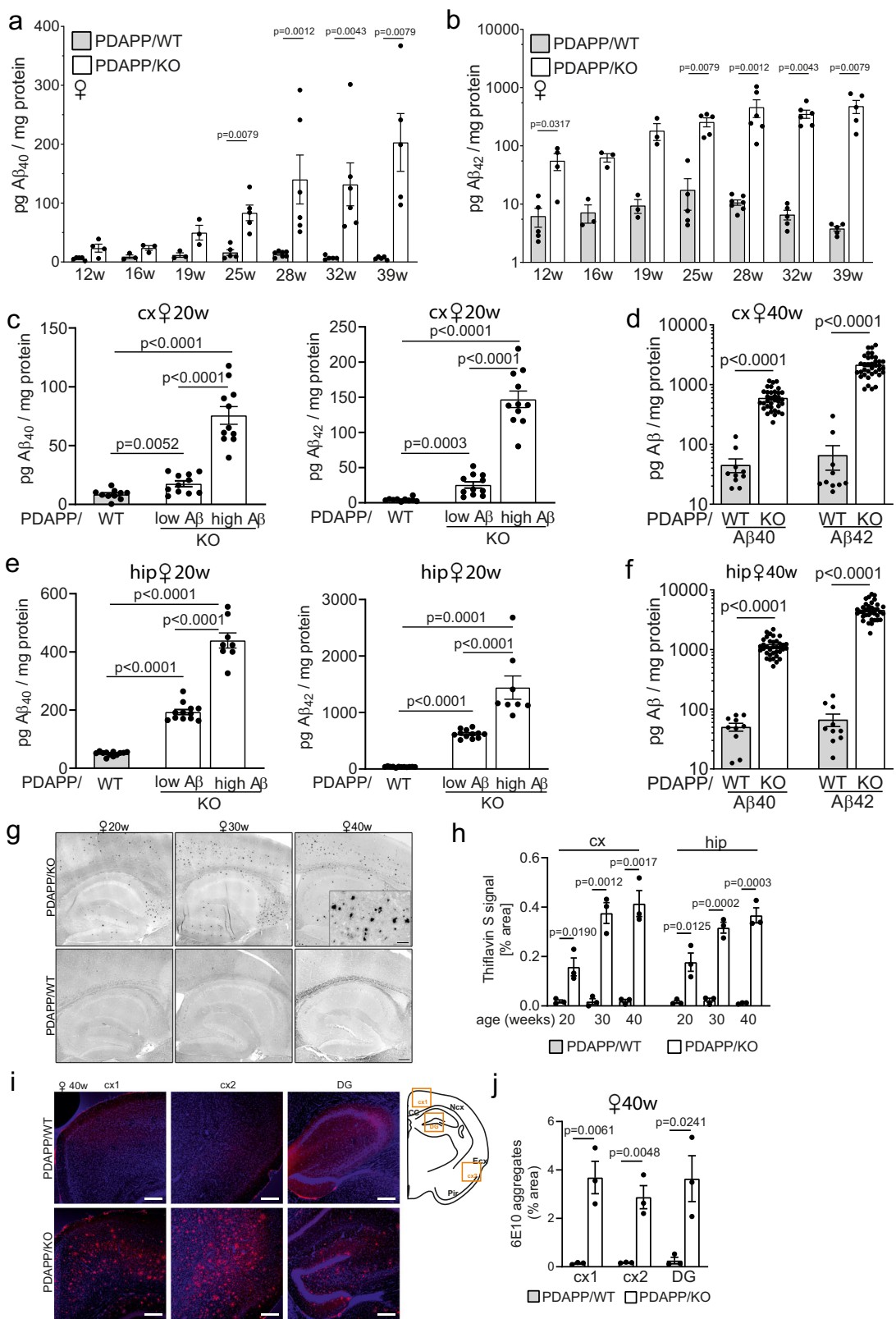

increased GFAP levels (Fig. 8d, e) and aggravated cell death (Fig. 8f) in hippocampi of aged PDAPP/asKO females. Also, levels of the early stress marker IL-33, already increased in PDAPP/KO mice at a young age (20 weeks), were significantly elevated in the hippocampi of aged PDAPP/asKO females (Fig. 8g). Contrary to the PDAPP/KO model, IL-33 levels (Fig. 8h), but also GFAP (Fig. 8d, e) and cell death (Fig. 8f), in PDAPP/asKO mice only increased in hippocampi, not cortices. These

observations were in agreement with elevated Aβ levels being restricted to hippocampi in the conditional mouse model.

## Aβ stress in astrocytes results in aberrant induction of δ secretase activity

Taken together, our studies in obligate and conditional SORCS2-deficient AD mouse models argued for loss of receptor activity in

**Fig. 2 | SORCS2 deficiency causes age-dependent brain amyloidosis in a mouse model of AD.** Age-dependent increase in soluble $A\beta_{40}$ (a) and $A\beta_{42}$ (b) levels in cortex of PDAPP/KO females as compared to PDAPP/WT controls. Data are given as mean ± SEM from $n = 3$ (16 w, 19 w.), $n = 4$ (12 w. KO), $n = 5$ (12 w. WT; 25w. WT and KO; 32 w. WT; 39 w. WT and KO), $n = 6$ (28 w. KO, 32 w. KO), $n = 7$ (28 w. WT) animals per group (two-sided unpaired Mann-Whitney U test). **c, d** Levels of soluble $A\beta_{40}$ and $A\beta_{42}$ in cortex (cx) of PDAPP/WT and PDAPP/KO mice at 20 (**c**) and 40 weeks of age (**d**). Data are given as mean ± SEM. $n = 10$ (**c**, WT, $A\beta_{40}$ and $A\beta_{42}$; **d**, WT, $A\beta_{40}$ and $A\beta_{42}$), $n = 11$ (**c**, $A\beta_{40}$ and $A\beta_{42}$, KO low and high; **d**, $A\beta_{40}$ and $A\beta_{42}$, WT), $n = 35$ (**d**, KO, $A\beta_{40}$ and $A\beta_{42}$) animals per group. Data were analyzed with two-sided unpaired Student's $t$-test for (**c**), and unpaired Mann-Whitney U test for (**d**). **e, f** Experiment as in (**c, d**) but testing hippocampal (hip) extracts from young (20 weeks, **e**) or aged (40 weeks, **f**) PDAPP/WT and PDAPP/KO females. Data are given as mean ± SEM from $n = 8$ (**e**, $A\beta_{40}$ and $A\beta_{42}$ KO high), $n = 12$ (**e**, $A\beta_{40}$ and $A\beta_{42}$, WT and KO low), $n = 10$ (**f**, $A\beta_{40}$ and $A\beta_{42}$ WT) $n = 35$ (**f**, $A\beta_{40}$ and $A\beta_{42}$, KO) animals per group (unpaired two-sided Student's $t$-test in panel **e**, unpaired two-sided Mann-Whitney U test in panel **f**). Note logarithmic scales in (**b, d**, and **f**). **g, h** Thioflavin S staining of sagittal brain sections documents plaque burden in hippocampal and cortical regions of 20, 30, or 40 weeks old PDAPP/WT and PDAPP/KO females. Representative images from a total of 3 animals per genotype are shown in (**g**). Scale bar: 200 μm (overview), 40 μm (zoom-in). Quantification of plaque burden is given in (**h**), expressed as % of total brain area analyzed. Data from $n = 3$ animals per genotype/timepoint are shown (mean ± SEM, unpaired two-sided Student's $t$-test). **i, j** In (**i**), representative images of cx and dentate gyrus (DG) sections from 40 weeks old female PDAPP/WT and PDAPP/KO mice immunostained for Aβ (antibody 6E10, red) are shown. Nuclei were counterstained with DAPI (blue). Brain areas analyzed are indicated in the schematic to the right. Scale bar: 200 μm. In (**j**), quantification of Aβ immunoreactivity from 3 animals per genotype is shown, expressed as % of total brain area (mean ± SEM, unpaired two-sided Student's $t$-test). $p$-values for all statistically significant differences are indicated on the graphs. Source data are provided in the Source Data file.

astrocytes to induce AD-like pathologies. Because these phenotypes required an initial trigger from the PDAPP transgene, we focused on amyloidogenic processing as a possible underlying cause of these pathologies. As more pronounced phenotypes of late-stage AD were seen in global as compared to astrocytic SORCS2 deficiency, which rather represents a model of early AD pathology, we carried out the subsequent studies in PDAPP/KO animals.

Initially, we measured levels of soluble (s) APPα and sAPPβ, cleavage products of APP processing by α and β secretases, respectively (Fig. 9a). Surprisingly, levels of sAPPα were reduced, but levels of sAPPβ were undetectable in cortex or hippocampus of 40 weeks old PDAPP/KO females when compared to PDAPP/WT animals (Fig. 9b). An explanation for this apparent paradox of massive production of Aβ, albeit at undetectable levels of sAPPβ, may stem from the activity of asparagine endopeptidase (AEP), also known as legumain (*Lgmn*) or δ secretase. Expression of δ secretase is induced during acute or chronic insults to the brain, as in AD, Parkinson's disease, stroke, or epilepsy[42–45]. The enzyme acts on APP, generating an extended stub (labeled CTFδ in Fig. 9a) that represents a favorable substrate for β secretase processing. Cleavage by δ secretase generates a shortened sAPP fragment (sAPPδ) that presumably lacks the recognition site for antibodies used in a commercial sAPPβ ELISA (www.mesoscale.com), providing an explanation for our inability to detect sAPPβ in PDAPP/KO mice. This assumption was supported by Western blotting showing a shorter soluble APP fragment with the expected molecular weight of sAPPδ in cortical (Fig. 9c, d) and hippocampal (Fig. 9e, f) extracts from aged PDAPP/KO females. Aggravated amyloidogenic processing was further substantiated by an increase in CTFβ stubs, and a concomitant depletion of mature APP from cortex (Fig. 9c, d) and hippocampus (Fig. 9e, f) of PDAPP/KO female mice.

Mass spectrometry analysis of recombinant APP preparations treated with PDAPP/KO brain extracts documented cleavage at N585, the site shown to produce sAPPδ[43], substantiating amyloidogenic processing by δ secretase in this tissue (Fig. 9g).

Of note, δ secretase activity also drives tau pathology as it proteolytically inactivates I2PP2A, an inhibitor of PPA2, the main enzyme responsible for dephosphorylation of tau[46–48]. Consequently, δ secretase actions result in tau hyperphosphorylation, accelerating AD pathology in mouse models double transgenic for human *APP* and *TAU*[49,50]. In addition, δ secretase has been shown to cleave tau, producing a fragment $tau_{N368}$ prone to aggregation[42]. In line with these facts, levels of cleaved $tau_{N368}$, as well as $tau_{N368}/tau_{total}$ ratio, were increased in 40 weeks old PDAPP/KO female mice compared to matched controls (Fig. 9h). No increase in levels of $tau_{N368}$ or $tau_{N368}/tau_{total}$ ratio were observed in aged PDAPP/asKO females (Supplementary Fig. 5e).

Increased levels of δ secretase as the likely cause of aberrant amyloidogenic processing and tau pathology in PDAPP/KO was supported by elevated levels of full-length (pro-form) and cleaved (active) forms of the enzyme in cortex (Fig. 9c-d) and hippocampus (Fig. 9e-f). No corresponding increases were seen for *Lgmn* transcript levels in cx and hip (Supplementary Fig. 6a) or isolated astrocytes or neurons (Supplementary Fig. 6b), suggesting a post-transcriptional mechanism to raise protein levels in the aged PDAPP/KO brain. Increased activity of δ secretase, as a consequence of elevated protein levels, in cortex and hippocampus of aged PDAPP/KO mice was directly confirmed using a home-made activity assay based on proteolytic cleavage of a fluorogenic enzyme substrate (Fig. 9i, see methods for details). Increased levels (Supplementary Fig. 6c) and activity (Supplementary Fig. 6d) of δ secretase were also seen in cortex and hippocampus of aged PDAPP/KO males. Contrary to the situation in aged PDAPP/KO mice, δ secretase activity was not increased in young PDAPP/KO female animals at 12 weeks of age (Supplementary Fig. 6e), indicating that induction of δ secretase activity in SORCS2-deficient brains aligns well with the appearance of profound amyloid and tau pathologies. In line with a less pronounced pathology in mice with astrocyte-specific *Sorcs2* inactivation, δ secretase levels were not significantly increased in aged PDAPP/asKO females compared with matched controls (Supplementary Fig. 6f). Importantly, neither levels of murine APP (Supplementary Fig. 6g) nor levels (Supplementary Fig. 6h) or activity (Supplementary Fig. 6i) of δ secretase were increased in aged KO female mice lacking the PDAPP transgene, again corroborating that SORCS2 deficiency alone is insufficient to induce δ secretase activity.

Expression of δ secretase increases with brain age[43] and is seen mainly in neurons in the AD brain[49]. To interrogate mechanisms whereby astrocyte distress increases δ secretase levels in the brain, we set up a co-culture model of primary astrocytes (WT or KO) and SH-SY5Y neuroblastoma cells (Fig. 10a). In this model, the co-cultures were treated with control media or with media conditioned with Aβ, and the consequential responses in δ secretase expression assayed. Remarkably, while neither levels of δ secretase protein nor transcript increased in WT or KO astrocytes when exposed to Aβ (Fig. 10b, c), SH-SY5Y cells responded to Aβ stress in the presence of KO astrocytes with a significant increase in δ secretase levels compared to co-culture with Aβ-stressed WT astrocytes (Fig. 10d). Similar to the situation in the murine brain, this increase in δ secretase protein was not reflected by a similar increase in transcript (Fig. 10e). Jointly, these findings support a molecular concept of heightened Aβ stress response in SORCS2-deficient astrocytes, that results in aberrant increases in δ secretase levels in neurons in trans.

## Discussion

Mechanisms defining the role of astrocytes in age-related dementia still remain poorly understood. Now, our data identified astrocyte distress as an important disease-promoting process at early stages of

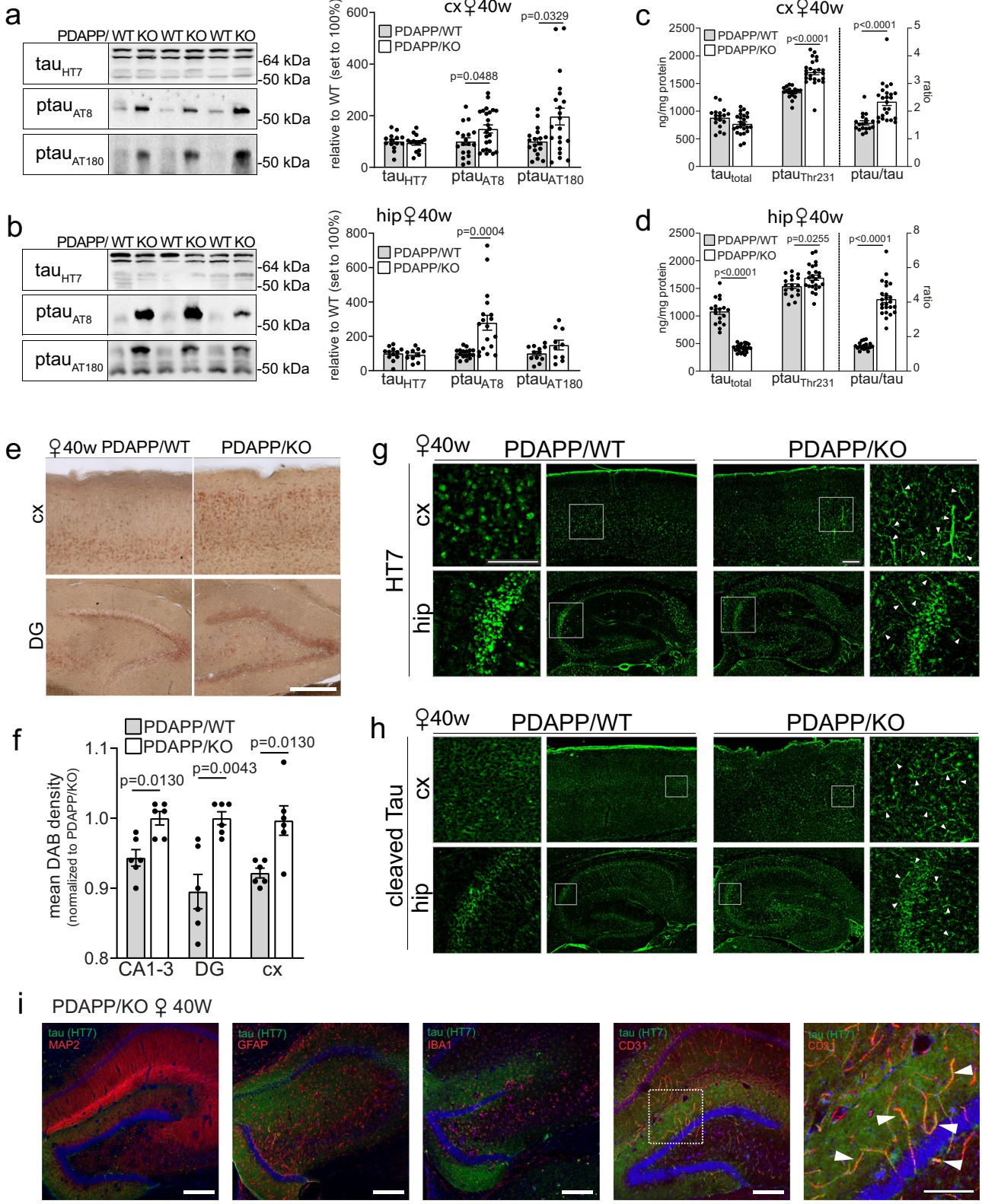

AD (Fig. 10f). According to our concept, an inability to cope with aggravated insults from Aβ results in astrocyte reactivity, triggering widespread gliosis and brain inflammation. These stress responses, in turn, induce δ secretase activity, accelerating amyloidogenic processing and tau pathology, and resulting in a vicious cycle that drives exhaustion and eventual death of astrocytes.

Amyloid plaques and tau hyperphosphorylation are two distinguishing features of AD in patients. A comorbidity, characterized by massive increase in Aβ and murine tau hyperphosphorylation, is seen in a mouse model of hypersensitivity of astrocytes to Aβ, induced by inactivation of the protective stress response gene *Sorcs2*. Increased astrocyte distress coincides with a pro-inflammatory brain milieu and with prominent gliosis, further characteristics of human AD. These phenotypes depend on a trigger from PDAPP. The extent of amyloid and tau comorbidities in PDAPP/KO mice is striking, given that *APP^Ind* is a rather subtle murine model of AD characterized by modest levels of

**Fig. 3 | SORCS2 deficiency induces tau pathology in a mouse model of AD. a, b** Representative Western blot analyses, and densitometric quantification of replicate blots thereof, document levels of total tau (HT7) as well as phosphorylated variants ptau_Ser202/Thr205 (AT8) and ptau_Thr231 (AT180) in cortical (**a**) and hippocampal (**b**) brain extracts of 40 weeks old PDAPP/WT or PDAPP/KO females. Data are expressed as relative to WT (set to 100%) and given as mean ± SEM from $n = 10$ (**b**, HT7 and AT180 KO), $n = 12$ (**b**, HT7 and AT180 WT), $n = 14$ (**a**, HT7 WT and KO), $n = 17$ (**b**, AT8 WT), $n = 18$ (**a**, WT AT8 and AT180; **b**, AT8 KO), $n = 21$ (**a**, AT180 KO), $n = 24$ (**a**, AT8, KO) animals per group (two-sided unpaired Student's $t$-test). Levels of total (tau_total) and ptau_Thr231, as well as ratio of ptau_Thr231/tau_total, in cortex (**c**) and hippocampus (**d**) of 40 weeks old female mice as determined by ELISA. Data are given as mean ± SEM from $n = 18$ (WT), $n = 24$ (KO) animals per group (two-sided unpaired Student's $t$-test). **e, f** Immunostaining for ptau_Ser202/Thr205 (AT8 antibody) on histological sections of cortical (cx) and hippocampal (hip) brain regions of 40 weeks old PDAPP/WT and PDAPP/KO female mice. Panel **e** depicts exemplary images of a total of six animals per genotype. Panel **f** gives the quantification of ptau_Ser202/Thr205 levels based on mean 3,3´-diaminobenzidine (DAB) intensities in

5 sections each from 6 animals per genotype. Data are given as mean ± SEM (two-sided unpaired Mann-Whitney U test). Scale bar, 200 µm. **g, h** Representative images from a total of 3 animals per genotype of cx and hip sections from 40 weeks old female PDAPP/WT and PDAPP/KO mice immunostained for total tau (**g**, HT7 antibody, green) or cleaved forms of tau_Asp421, Asp422 (**h**, tauC3, green). Images are given as overview images and higher magnification zoom-in (indicated as white squares in the overviews). Arrowheads indicate fibrillary appearance of tau and tauC3 immunoreactivity in PDAPP/KO tissue. Scale bar: 150 µm (inset), 200 µm (overview). **i** Co-immunostaining of total tau (HT7, green) with markers of endothelial cells (CD31, red), neurons (MAP2, red), astrocytes (GFAP, red) or microglia (IBA1, red) on hippocampal brain sections from 40 weeks old PDAPP/KO female mice. Nuclei were counterstained with DAPI (blue). White box in the CD31 image indicates the area magnified in the panel given to the right. White arrowheads in the inset indicate the colocalization of tau with CD31. Scale bar, 200 µm, inset: 100 µm. The staining was done for 3 animals per group with similar results. $p$-values for all statistically significant differences are indicated on the graphs. Source data are provided in the Source Data file.

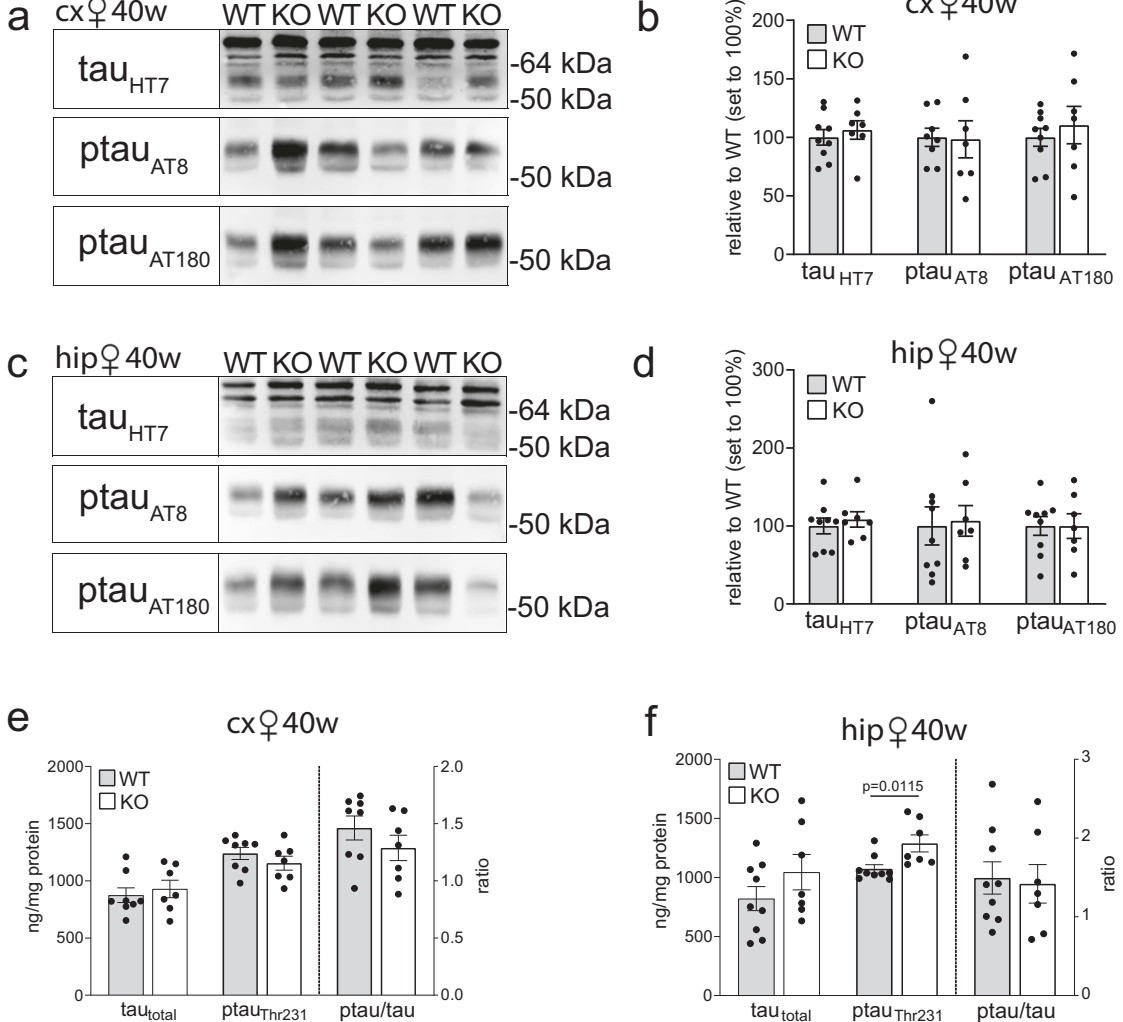

**Fig. 4 | Tau pathology in SORCS2-deficient female mice requires Aβ trigger. a, b** Representative Western blot analyses, and densitometric quantification of replicate blots thereof, document levels of total tau (HT7) as well as phosphorylated variants ptau_Ser202/Thr205 (AT8) and ptau_Thr231 (AT180) in cortical extracts of 40 weeks old WT and KO female animals lacking the PDAPP transgene. Data are expressed as relative to WT (set to 100%) and given as mean ± SEM from $n = 7$ (KO), $n = 8$ (AT8 WT), $n = 9$ (HT7 and AT180 WT) animals per group (two-sided unpaired Mann-

Whitney U test). **c, d** Data as in (**a**) and (**b**) but for hippocampal extracts. $n = 7$ (KO), $n = 9$ (WT) **e, f** Levels of total (tau_total) and phosphorylated variants (ptau_Thr231) of tau, as well as ratio of ptau_Thr231/tau_total, in cx (**e**) and hip (**f**) of 40 weeks old WT and KO females. Data are given as mean ± SEM from $n = 7$ (KO **e, f**), $n = 8$ (**e**, WT), $n = 9$ (**f**, WT) animals per genotype (two-sided unpaired Mann-Whitney U test). $p$-values for all statistically significant differences are indicated on the graphs. Source data are provided in the Source Data file.

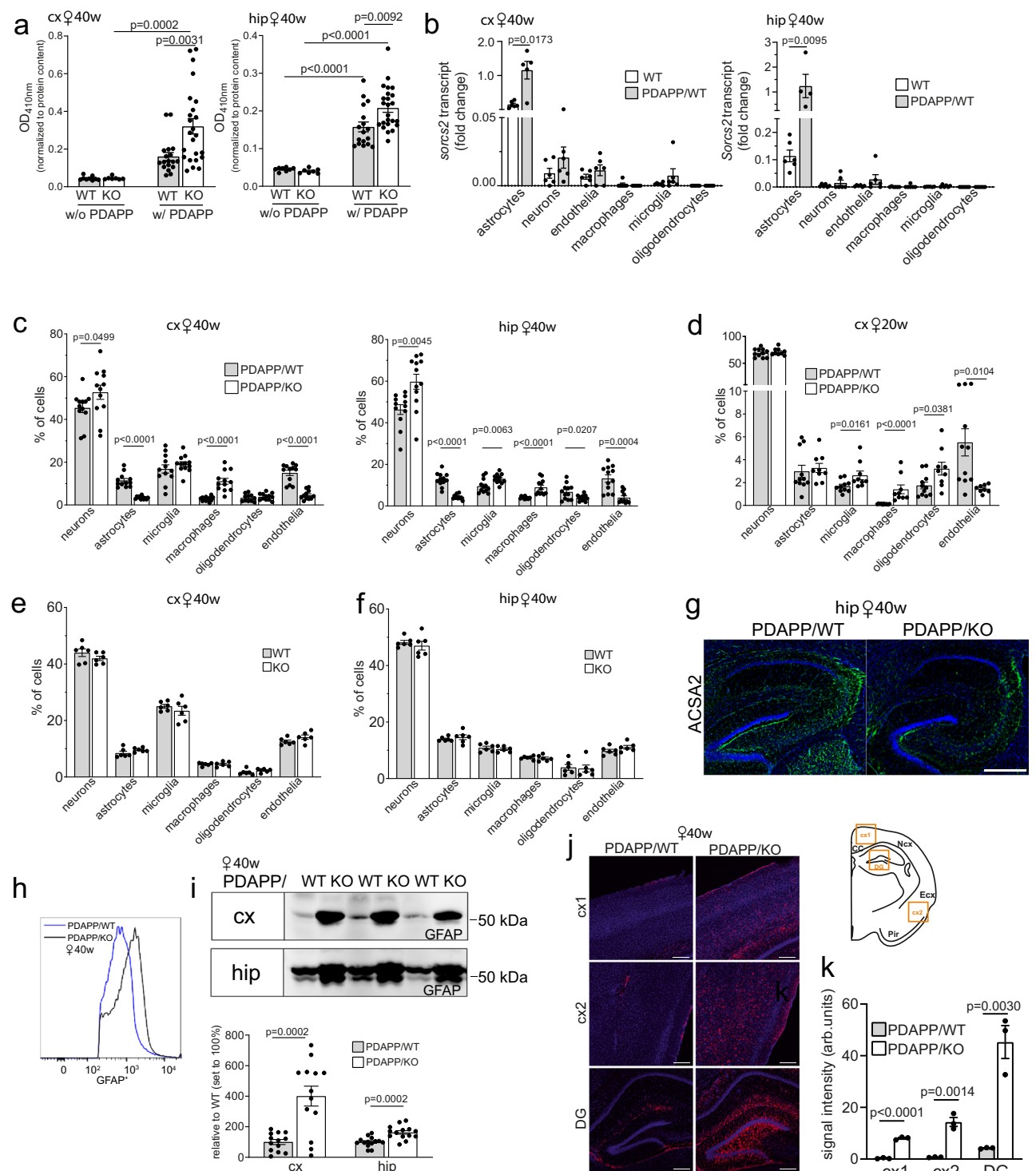

human Aβ and limited plaque deposition visible only after 9 months of age[22]. Thus, aggravation of phenotypes with a 30-fold increase in Aβ$_{42}$ levels and tau hyperphosphorylation, in the absence of a human *TAU* transgene, clearly stems from the lack of protective SORCS2 actions.

The temporal succession of phenotypic features seen with various mouse genotypes and ages establishes the causality of pathological processes caused by astrocyte dysfunction in our AD model. Pro-inflammatory activation of glia, as evidenced by cytokine profiling, is a phenotype already obvious in young PDAPP/KO mice with low Aβ burden (Supplementary Table 1). A relative increase in immune cell types (Fig. 5d), with a shift from microglia to macrophages (Fig. 7b) corroborates a pro-inflammatory milieu in young PDAPP/KO brains as

an early consequence of SORCS2 deficiency, preceding full-blown amyloid and tau pathologies at later stages. This conclusion is supported by increase in IL-33, an early indicator of cell stress already in PDAPP/KO mice with moderate Aβ burden (Fig. 8g). Increased Aβ levels are also seen in aged mice with astrocytic *Sorcs2* inactivation (Fig. 8b). This phenotype is less pronounced than in the obligate PDAPP/KO model, and restricted to the hippocampus, likely due to technical limitations of conditional receptor depletion. As tamoxifen-induced Cre-mediated *Sorcs2* gene disruption is limited temporally, mutant astrocytes will be continuously replaced by non-targeted cells, gradually reducing the phenotypic outcome of astrocytic loss of SORCS2. Obviously, our data do not exclude confounding effects from

**Fig. 5 | Aβ induces astrogliosis and loss of astrocytes in SORCS2-deficient mice. a** Nucleosome fragmentation assay documenting cell death in cortical (cx, left panel) and hippocampal (hip, right panel) brain extracts of 40 weeks old PDAPP/WT and PDAPP/KO females as well as WT and KO controls lacking PDAPP. Data are given as mean ± SEM from $n = 7$ (cx and hip, KO w/o PDAPP), $n = 8$ (hip, WT w/o PDAPP), $n = 9$ (cx, WT w/o PDAPP), $n = 17$ (hip, WT w/PDAPP), $n = 18$ (cx, WT w/ PDAPP), $n = 24$ (cx and hip, KO w/PDAPP) animals per group (ordinary two-way ANOVA with Tukey's multiple comparisons test). **b** Transcript levels for *Sorcs2* in the indicated brain cell types isolated by FACS from cx (left panel) and hip (right panel) of WT animals, with and without PDAPP transgene. Data are given as mean ± SEM from $n = 4$ (hip, w/PDAPP, astrocytes), $n = 5$ (cx, w/PDAPP, astrocytes), $n = 6$ (all other groups) animals per group (two-sided unpaired Mann-Whitney U test). **c** Cell type distribution in cx (left panel) and hip (right panel) of 40 weeks old PDAPP/WT and PDAPP/KO females as determined by FACS. Data are given as mean ± SEM from $n = 11$ (cx and hip, astrocytes KO; cx oligo WT; cx and hip, endothelia KO; hip, macrophages WT), $n = 12$ (all other groups) animals per group (two-sided unpaired Student's *t*-test). **d** Cell type distribution in cx of 20 weeks old PDAPP/WT and PDAPP/KO females based on FACS. Data are given as mean ± SEM from $n = 8$ (endothelia, KO), $n = 9$ (all other cell types, KO), $n = 10$ (microglia and macrophages, WT), $n = 11$ (neurons, astrocytes, oligo, endothelia, WT) animals per group (two-sided unpaired Mann-Whitney U test). FACS documenting relative distribution of various cell types in cx (**e**) and hip (**f**) of 40 weeks old WT and KO females lacking PDAPP. Data are given as mean ± SEM from $n = 6$ animals per genotype (two-sided unpaired Mann-Whitney U test). **g** Immunostaining of hippocampal regions of 40 weeks old PDAPP/WT and PDAPP/KO females for astrocyte marker ACSA2 (green). Nuclei were counterstained with DAPI (blue). Scale bar, 400 μm. The staining was performed for 3 animals per group with similar results. **h** Representative FACS histogram of GFAP levels in sorted astrocytes from aged PDAPP/WT and PDAPP/KO female brains. **i** Representative Western blot for GFAP in brain extracts (upper panel), and quantification from densitometric scanning of replicate blots (lower panel), evaluating its levels in cx and hip of 40 weeks old PDAPP/KO females and matched PDAPP/WT controls. Data are expressed as relative to WT (set to 100%) and given as mean ± SEM from $n = 13$ (cx), $n = 14$ (hip) animals per group (two-sided unpaired Student's *t*-test). **j** Representative images of GFAP immunostaining (red) in brain regions of 40 weeks old PDAPP/KO as compared to PDAPP/WT female mice. A schematic depicting analyzed cortical (cx1, cx2) and dentate gyrus (DG) regions is shown. Nuclei were counterstained with DAPI (blue). Scale bar: 200 μm. **k** GFAP signal intensities (arbitrary units, arb. units) quantified from replicate immunostainings exemplified in (**j**). Data are given as mean ± SEM from $n = 3$ animals per genotype (two-sided unpaired Student's *t*-test). *p*-values for all statistically significant differences are indicated on the graphs. Source data are provided in the Source Data file.

neuronal loss of SORCS2 expression on AD pathology in the PDAPP/KO mouse model. However, an increased Aβ burden as well as additional phenotypes, such as elevated levels of GFAP and IL-33, and induced cell death, shared by PDAPP mice with global and astrocyte-specific *Sorcs2* gene defect, argue that astrocytic SORCS2 deficiency plays a decisive role in AD pathology. While we cannot confidently compare levels of SORCS2 expression in astrocytes versus neurons based on transcripts in sorted cells or immunohistology in brain tissue, additional data strongly support a decisive role for astrocytic SORCS2 in AD. This conclusion is based on receptor transcript levels to specifically increase in astrocytes, but not neurons, in response to the PDAPP transgene (Fig. 5b). Also Aβ-induced cell loss is seen in astrocytes, but not neuronal cell types, in PDAPP/KO mice (Fig. 5c).

Prior studies have reported the sensitivity of astrocytes to Aβ exposure[4], and a corresponding astrocyte cell death in AD patients' brains[51–53]. Astrocytes participate in removal of amyloid from the brain parenchyma[54,55]. However, contrary to other cell types, such as microglia, astrocytes accumulate rather than degrade ingested material[56,57]. As a consequence, they acquire a high intracellular load of toxic amyloid that impacts endo-lysosomal function and energy homeostasis[3,56,58]. Exposure to Aβ also triggers release of pro-inflammatory cytokines and chemokines from astrocytes[10,59,60], evoking inflammation that further enhances brain Aβ production[61–63] and tau pathology[62]. SORCS2-deficient astrocytes exhibit increased uptake and intracellular accumulation of Aβ (Fig. 1b, d). Amyloid buildup coincides with astrocyte reactivity (Fig. 1e), defects in lysosomal acidification (Fig. 1f), and apoptotic cell death (Fig. 1g, i). These findings document a role for SORCS2 in preventing excessive uptake and/or accumulation of Aβ in astrocytes. Enhanced clearance of Aβ by SORCS2-deficient astrocytes is supported by reduced Aβ levels initially seen in the brains of young PDAPP /asKO animals before progressing pathology increases amyloid burden (Fig. 8a).

SORCS2 is an intracellular sorting receptor that acts in protective stress responses in multiple tissues. Noteworthy is its mode of action in trans, whereby expression in supportive cell types, such as astrocytes or pancreatic α-cells, triggers release of factors that provide protective actions to functional cell types, such as endothelial cells in stroke[20] or β-cells in glucose stress[18]. Remarkably, similar mechanism seems to be operable in the AD brain, as loss of protective SORCS2 action in astrocytes triggers an aberrant response in stressed neurons, that includes increasing levels of δ secretase (Fig. 10d). The increase in δ secretase protein, seen both in the PDAPP brain and in SH-SY5Y cells in response to Aβ, is not correlated with a corresponding increase in

*Lgmn* transcript. This fact argues for a post-transcriptional mechanism to raise δ secretase levels and/or activity in neurons. Such mechanisms may entail expression control by phosphorylation[64].

Given the multifunctionality of SORCS2[12], proteins sorted by this receptor in astrocytes in response to Aβ exposure are difficult to predict, but may include clearance receptors for Aβ. One possible candidate is the neurotrophin receptor p75NTR, an established interacting partner of SORCS2[13,16] implicated in Aβ binding and uptake[65,66]. Potentially, SORCS2-dependent sorting of p75NTR reduces Aβ ingestion, protecting astrocytes from amyloid overload. Overload, in turn, may impair the release of protective factors that normally prevent pathological responses in the AD brain, including neuronal δ secretase expression. In line with this hypothesis, the closely related VPS10P domain receptor SORCS3 was shown to target p75NTR to lysosomes to promote its degradation[67]. Whatever the protective function of SORCS2 in the context of AD may be, it is universal to the entire astrocyte lineage as concluded from loss of all tested astrocyte subtypes in the PDAPP/KO brain (Supplementary Fig. 3d).

Astrocyte reactivity has been proposed to enhance amyloid burden, yet the underlying molecular mechanism remained elusive (reviewed in ref. 68). Our findings shed light on this process by implicating a potential culprit, δ secretase. This endo-lysosomal peptidase cleaves APP at N585 and N373, accelerating Aβ production by generating a favored β secretase substrate[43,50]. Remarkably, δ secretase activity also drives tau pathology through cleavage of I2PP2A[46–48] and production of tau$_{N368}$[42]. With relevance to our hypothesis, δ secretase expression and activity in the brain is induced by inflammation[69,70], a phenotype prevalent in the SORCS2-deficient AD brain. While we have not formally documented rescue of PDAPP/KO phenotypes by inactivation of *Lgmn* in vivo, our findings strongly implicate induction of neuronal δ secretase activity as a molecular mechanism whereby astrocyte stress provides an early trigger for amyloid and tau comorbidities in the AD brain.

## Methods
This research complies with all relevant ethical regulations. Animal experiments were conducted in full compliance with national legislation and European regulations (Directive 2010/63/EU of the European Parliament and of the Council of 22 September 2010). Experimental protocols were approved by the authorities of the State of Berlin (animal protocol approval numbers X9007/17; X9009/22; G0105/22) and the First Ethical Committee in Warsaw (animal protocol approval number 1375P1/2022).

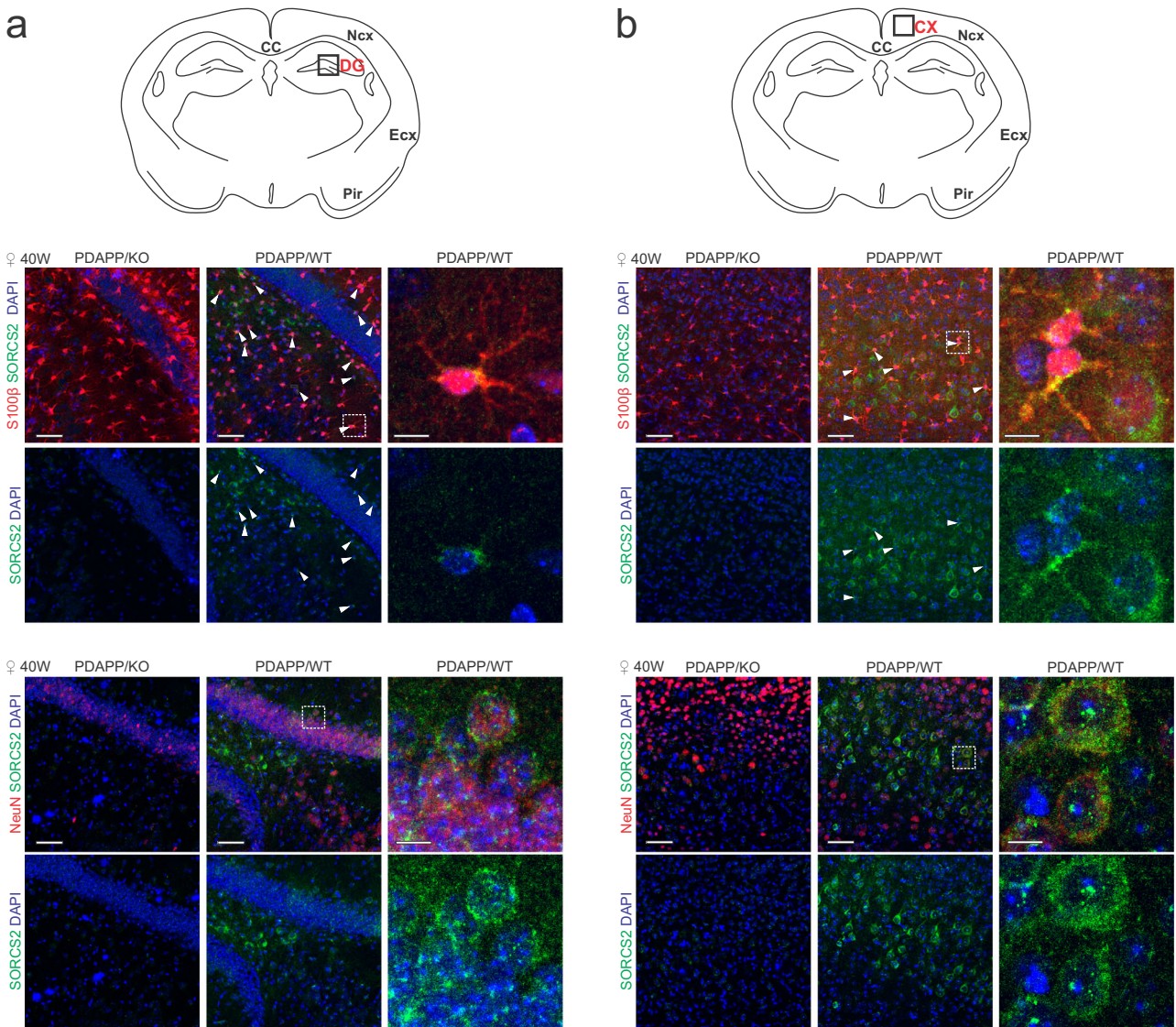

**Fig. 6 | SORCS2 is present both in neurons and astrocytes in aged PDAPP mice.** Representative images from the total of 3 animals per genotype of dentate gyrus (**a**) and cortex (**b**) in the brain sections from 40 weeks old female PDAPP/WT and PDAPP/KO mice immunostained for SORCS2 (green) and markers of astrocytes (S100β, upper panels, red) and neurons (NeuN, lower panels, red). DAPI was used to stain cell nuclei (blue). Brain regions imaged in panels (**a**) and (**b**) are marked in the schemes above. Arrowheads point to SORCS2-positive astrocytes. White boxes in the PDAPP/WT images indicate the area magnified in the panels given to the right. Scale bar: 50 μm, in insets: 10 μm.

## Human brain specimens

Brain autopsy specimens from the frontal cortex of AD patients and control subjects were obtained from the Netherlands Brain Bank (Netherlands Institute for Neuroscience, Amsterdam) and the MRC London Brain Bank for Neurodegenerative Diseases (Institute of Psychiatry, King's College London). The ethnicity of samples was white. All material was collected from donors for or from whom a written informed consent for a brain autopsy and the use of the material and clinical information for research purposes had been obtained by the Netherlands Brain Bank or the MRC London Brain Bank. Detailed personal information, including age, gender, neuropathological stage, *APOE* genotype, of the individuals were provided by the brain banks. RNA was extracted from the snap-frozen tissue specimens by standard procedures and used for quantitative RT-PCR.

## Mouse models

Mice with targeted disruption of *Sorcs2* (KO) have been described before ref. 16. For analysis of human APP processing, wildtype (WT)

and KO mice were crossed with the PDAPP line 109[22]. All animals were kept on an inbred C57BL/6 N background and studied at 20 (young cohort) or 38-40 (old cohort) weeks of age. Astrocyte- and neuron-specific *Sorcs2* KO lines were generated by crossing the *Sorcs2*[lox/lox] line[15] with Cre transgenic strains Aldh1l1-Cre (JAX: #023748) or BAF53b-Cre (JAX: #027826). Subsequently, both lines were bred to PDAPP. Astrocyte-specific *Sorcs2* inactivation was induced in *PDAPP/Aldh1l1-Cre/Sorcs2*[lox/lox] animals by injection with 100 mg/kg body weight of tamoxifen (Sigma #T5648-1G) for five consecutive days at 10 – 12 weeks of age. Mice were sacrificed by cervical dislocation or by transcardial perfusion, following sedation using ketamine/xylazine or pentobarbital in line with local guidelines and approved protocols. All animal experimentations were performed according to institutional guidelines following approval by the authorities of the State of Berlin (animal protocol approval numbers X9007/17; X9009/22; G0105/22) or the First Ethical Committee Warsaw (animal protocol approval number 1375P1/2022). The animals were kept on normal chow, 12 h/ 12 h dark/light cycle, in stable temperature (22 °C +/− 2 °C) and

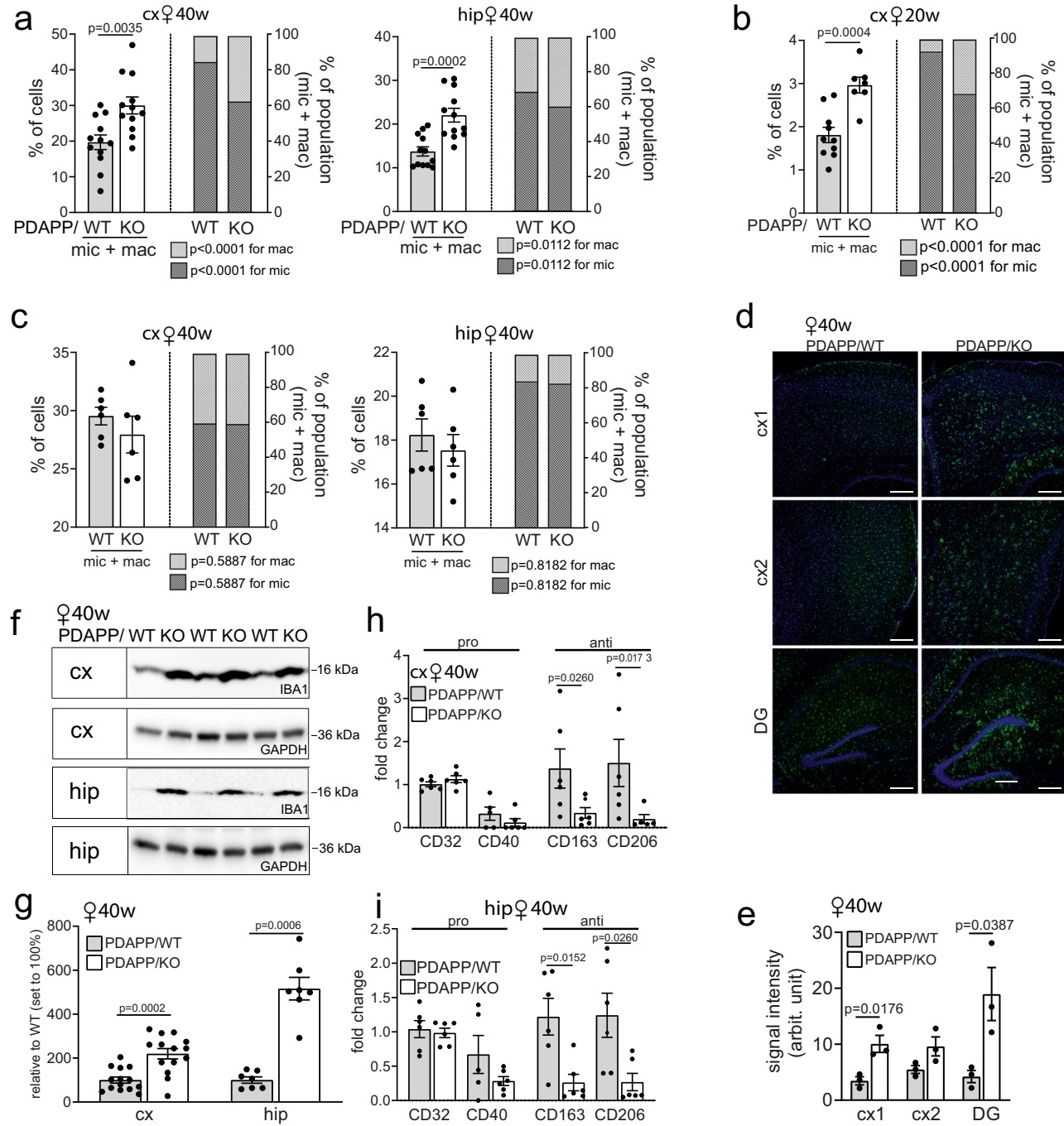

**Fig. 7 | Enhanced pro-inflammatory microglia response in PDAPP mice lacking SORCS2. a** Quantification of cell numbers of microglia and macrophages in cortex (cx) and hippocampus (hip) of 40 weeks old PDAPP/WT and PDAPP/KO mice using FACS. Data are expressed as % of the total number of cells (left axis) or as % of the microglia and macrophage subpopulation (right axis), and are given as mean ± SEM from $n = 11$ (cx, KO, %mac and %mic), $n = 12$ (all other groups) animals per group (two-sided unpaired Student's *t*-test). **b** Experiment as in (**a**), but quantifying the number of microglia and macrophages in cx of 20 weeks old PDAPP/WT and PDAPP/KO female mice. Data are given as mean ± SEM from $n = 7$ (KO mic+mac), $n = 9$ (KO, %mac and %mic), $n = 10$ (WT mic+mac), $n = 11$ (WT, %mac and %mic) animals per group (two-sided unpaired Mann-Whitney U test). **c** Experiments as in (**a**), but using brain samples from 40 weeks old WT and KO mice lacking PDAPP. Data are given as mean ± SEM from $n = 6$ animals per genotype (two-sided unpaired Mann-Whitney U test). **d** Representative immunofluorescence images of cortical regions (cx1, cx2) and the dentate gyrus (DG) of PDAPP/KO and PDAPP/WT females

stained for IBA1 (green). Nuclei were counterstained with DAPI (blue). Scale bar: 200 μm. **e** Signal intensities for IBA1 (arbitrary units, arb. units), as determined by immunostainings in (**d**). Data are given as mean ± SEM from $n = 3$ animals per genotype (two-sided unpaired Student's *t*-test). **f, g** IBA1 in cx and hip lysates, as determined by Western blot (**f**), and densitometric scanning of replicate blots (**g**), documenting its levels in aged PDAPP/WT and PDAPP/KO females. Data are expressed as relative to WT (set to 100%) and given as mean ± SEM from $n = 7$ (hip), $n = 14$ (cx) animals per group (two-sided unpaired Student's *t*-test for cortex, two-sided unpaired Mann-Whitney U test for hippocampus). Transcript levels for markers of pro- or anti-inflammatory responses in microglia sorted from cx (**h**) and hip (**i**) of 40 weeks old PDAPP/WT and PDAPP/KO female mice. Data are expressed as relative to WT (set to 1) and given as mean ± SEM for $n = 5$ (**h**, WT, CD40; KO, CD206; **i**, WT, CD40), $n = 6$ (all other groups) animals per group (two-sided unpaired Mann-Whitney U test). *p*-values for all statistically significant differences are indicated on the graphs. Source data are provided in the Source Data file.

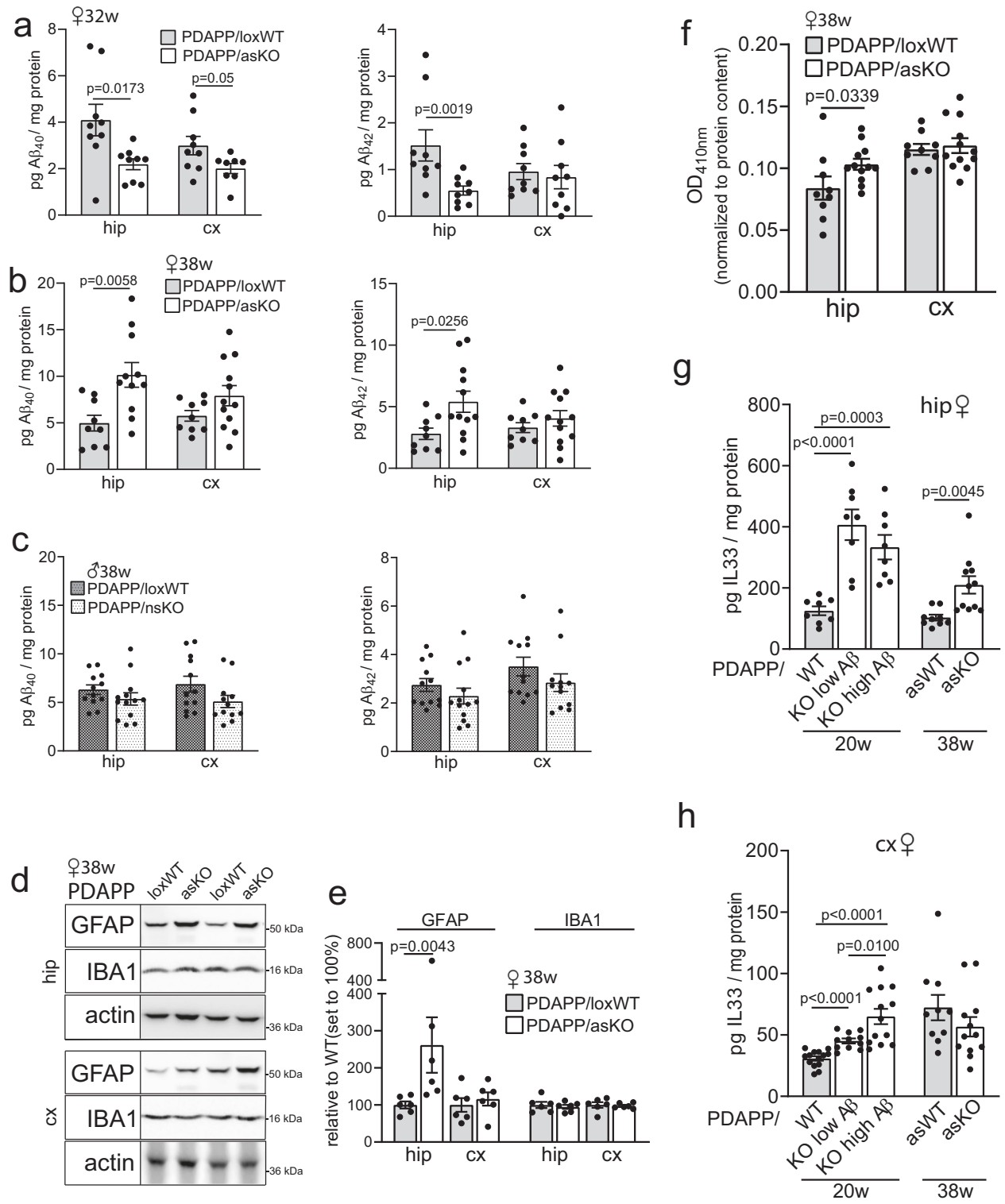

humidity (55% +/− 10%). Sex of the mice was considered in the study design.

**Analysis in mouse tissues**

Expression analyses in mouse tissues by quantitative RT-PCR, Western blotting, ELISA, or immunohistology were performed using standard procedures as detailed in the sections below. Uncropped and unprocessed scans of Western blots results are provided in the Source Data file. Information on RT-PCR primers and probes is given in the section 'Quantitative RT-PCR' below. Cell death in brain tissues was determined as the amount of cytoplasmic histone-associated DNA fragments (mono- and oligonucleosomes) using the Cell Death Detection ELISA (Roche #11544675001) according to the manufacturer´s protocol. For δ secretase activity assays, brain tissues were homogenized in assay buffer (20 mM citric acid, 60 mM $Na_2HPO_4$, 1 mM DTT, 1 mM EDTA, 0.1% CHAPS, 0.5% Triton X-100, pH 4.5) and the protein concentrations adjusted to 0.5 µg/µl using the same buffer. Extracts (100 µl) were mixed with 100 µl of 20 µM δ secretase Z-AAN-AMC substrate (Bachem #4033201.0050) and the enzyme activity determined by fluorimetry (Ex: 346 nm, Em: 443 nm) over 2 h.

**Fig. 8 | Astrocyte-specific inactivation of *Sorcs2* increases Aβ levels and induces cell stress in the PDAPP mouse brain. a** Levels of soluble $A\beta_{40}$ and $A\beta_{42}$ in hippocampus (hip) and cortex (cx) of 32 weeks old PDAPP animals (20 weeks post tamoxifen injection) homozygous for *Sorcs2*$^{lox/lox}$ (PDAPP/loxWT) or carrying an astrocyte-specific *Sorcs2* gene defect (PDAPP/asKO). Data are given as mean ± SEM from $n = 8$ ($A\beta_{40}$, asKO), $n = 9$ (all other groups) animals per group (two-sided unpaired Student's *t*-test). **b** Data as in (**a**), but from hip and cx extracts of 38 weeks old PDAPP/loxWT and PDAPP/asKO animals. Data are given as mean ± SEM for $n = 9$ (loxWT), $n = 11$ (hip, $A\beta_{42}$, asKO) $n = 12$ (other groups) animals per group (two-sided unpaired Student's *t*-test). **c** Levels of soluble $A\beta_{40}$ and $A\beta_{42}$ in cx and hip of 38 weeks old male mice, either PDAPP/loxWT or carrying a neuron-specific *Sorcs2* gene defect (PDAPP/nsKO). Data are given as mean ± SEM for $n = 13$ ($A\beta_{40}$ and $A\beta_{42}$, hip nsKO), $n = 12$ (other groups) animals per group (two-sided unpaired Mann-Whitney U test). Representative Western blot for GFAP and IBA1 in hip (**d**, upper panel) or cx brain extracts (**d**, lower panel), and quantification from densitometric scanning of replicate blots thereof (**e**), are shown. Data are expressed as relative to WT (set to 100%) and given as mean ± SEM from $n = 6$ animals per genotype (two-sided unpaired Mann-Whitney U test). **f** Nucleosome fragmentation assay documenting cell death in hip extracts of 38 weeks old PDAPP/asKO females and matched PDAPP/WT. Data are given as mean ± SEM from $n = 9$ (loxWT), $n = 12$ (asKO) animals per group (two-sided unpaired Mann-Whitney U test). Levels of IL-33 in hip (**g**) and cx (**h**) extracts of PDAPP/WT or PDAPP/KO female mice at 38 weeks of age. Young females are separated into two groups based on low versus high brain Aβ levels (cut off: 300 pg/ml $A\beta_{40}$ and 800 pg/ml $A\beta_{42}$ for hip; 35 pg/ml $A\beta_{40}$ and 60 pg/ml $A\beta_{42}$ for cx). Data are given as mean ± SEM from $n = 8$ (**g**, 20 w.), $n = 9$ (**g**, 38 w. loxWT), $n = 10$ (**h**, 38 w. loxWT), $n = 11$ (**h**, 38 w. asKO; h, 20 w. KO low), $n = 12$ (**h**, 38 w. asKO), $n = 13$ (**h**, 20 w. KO high), $n = 14$ (**h**, 20 w. WT) animals per group (two-sided unpaired Student's *t*-test). *p*-values for all statistically significant differences are indicated on the graphs. Source data are provided in the Source Data file.

## Flow cytometric analysis and quantification of brain cell types

Characterization of individual brain cell types was performed by FACS, following isolation of cells from adult mouse brains by Percoll gradients. In detail, brain regions were dissected, gently minced, and resuspended in Hanks´ Balanced Salt Solution (HBSS, Gibco #14185-045) containing 8 units of papain (Worthington Biochemical #LK003172) and 250 units of DNase I (Invitrogen #18047-019). Resuspended tissues were incubated in a water bath at 37 °C for 20 min and gently titrated 10-fold with a 20 G syringe halfway through the incubation time. After incubation, 5 ml of wash buffer (HBSS with 2 mM EDTA and 0.5% BSA) were added to the homogenates and filtered through a 70 µm mesh before cells were pelleted at 350 x *g* for 5 min. Cell pellets were resuspended in 900 µl PBS with 0.025% BSA and 100 µl Myelin Removal Beads II (Miltenyi Biotec #130-096-433) and incubated for 15 min at 4 °C. Then, cells were washed and resuspended in 2 ml of wash buffer. One ml of each suspension was placed on a LS column on the magnetic field separator. The flow through (all cells without myelinated oligodendrocytes) was collected. The LS column was washed with 4 ml wash buffer, collected in the same tube. These steps were repeated with another 1 ml aliquot of the same homogenate and both cell fractions were combined. To isolate oligodendrocytes, the LS columns were removed from the magnetic field separator and additionally loaded with 3 ml of wash buffer. The flow through was collected and the myelinated oligodendrocytes were pelleted at 350 x *g* for 5 min. The resulting pellets were resuspended in 5 ml of PBS with 20% isotonic Percoll PLUS (Millipore #E0414-250 ml) and centrifuged in swingout buckets at 310 x *g* for 20 min to remove the myelin sheath from the oligodendrocytes. The resulting pellets were washed to remove traces of Percoll PLUS, centrifuged, and combined with the main single cell suspensions. Cells were centrifuged again, resuspended in 100 µl wash buffer containing 1:100 Mouse BD Fc Block (BD Biosciences #553141), and transferred to reaction tubes. After 15–20 min of incubation at 4 °C, 100 µl of wash buffer were added containing antibodies directed against markers of microglia and macrophages (CD45-BV421; 1:200, BD Biosciences #563890), endothelia (CD49a-FITC; 1:200, Miltenyi Biotec #130-107-636), oligodendrocytes (O4-PE; 1:200, Miltenyi Biotec #130-117-357) and astrocytes (ACSA2-APC, 1:100, Miltenyi Biotec #130-116-245; S100b-PE, 1:100, Novus Bio #NBP2- 45267; GFAP-BV421, 1:100, Biolegend #644710; Aldh1l1-FITC, 1:100, Novus Bio #NBP2-50045F; SOX9-488 (1:100, #CL488-67439 proteintech), followed by incubation overnight at 4 °C. Prior to staining for GFAP-BV421, SOX9-488 and Aldh1l1-FITC, cells were fixed and permeabilized with Phosflow lyse/fix (BD Biosciences # 558049) and Perm Buffer III (BD Biosciences #558050) according to the manufacturers' instructions. The next day, stained cells were washed and FACS sorted using BD Aria Fusion. The FlowJo 10 Software was used for quantification of cell type numbers.

## Assessing the response of primary astrocytes to Aβ

Astrocytes were isolated from newborn or adult mouse brain as described in the section 'Primary astrocyte cultures' below. To determine Aβ uptake, primary astrocytes were treated with conditioned media from parental SY5Y cells (obtained from ATCC) or cell clone SY5Y-A, constitutively overexpressing human APP[69][21]. After 24–48 h of incubation, levels of Aβ in astrocyte supernatants and lysates were measured by ELISA (Meso Scale Discovery). To determine viability, astrocytes were seeded in 8 technical replicates at a density of 5000 cells per well in 96-well plates. Twenty-four hours after seeding, the medium was changed to conditioned media from cell lines SY5Y and SY5Y-A for 24 or 48 h. Finally, the medium was changed again to DMEM/HAMS F12 complete medium containing 1x Presto Blue reagent (Invitrogen #A13261). The fluorescence signal was measured after 60 min of incubation at excitation/emission wavelengths of 535/595 nm and ratio of (Aβ+) to (Aβ-) signals was calculated for each replicate.

For analysis of the lysosomal compartment, astrocytes were seeded at a density of 5000–7000 cells per well in 96-well plates for 24 h. Then, the medium was switched to conditioned media from cell lines SY5Y and SY5Y-A for 24 h. Then, cells were live-stained for 20 min with 50 nM LysoTracker Red DND-99 (L7528; Thermo Fisher Scientific) and Hoechst and scanned using Opera Phenix high content screening microscope (PerkinElmer) with 40 × 1.1 NA water immersion objective. The Harmony 4.9 software (PerkinElmer) was used for image acquisition and quantitative analysis. More than 10 microscopic fields were analysed for each experimental condition to quantify lysosome number and cumulative intensity of their fluorescence. The average number of cells used for analysis was between 350 and 450 per condition. Maximum intensity projection images were obtained from three to five Z-stack planes with 1 µm interval. Images were assembled in ImageJ and Photoshop (Adobe) with only linear adjustments of contrast and brightness.

## Expression analyses in mouse tissues and primary cell types

Protein extracts from cortical or hippocampal brain tissues were generated using standard protocols. All steps were performed at 4 °C using solutions containing protease (Complete #11836145001, Roche) and phosphatase (PhosStop 04906837001, Roche) inhibitors. Soluble and membrane protein fractions were obtained by homogenization of brain samples in solution A (20 mM Tris-HCl, 2 mM MgCl2, 250 mM sucrose, pH 7.5) using a pestle B (10 strokes). After centrifugation at 1000 x *g* for 5 min, the supernatants were transferred to an ultra-centrifuge tube. The remaining pellet was again homogenized in solution A and centrifuged, and both supernatants combined. Soluble protein fractions were obtained by ultracentrifugation of the combined supernatants at 175,000 x *g* for 30 min. The remaining pellets were lysed in lysis buffer (20 mM Tris-HCl, 10 mM EDTA, 1% NP40, 1% Triton X-100, pH 7.4) to derive the membrane protein fraction. Total

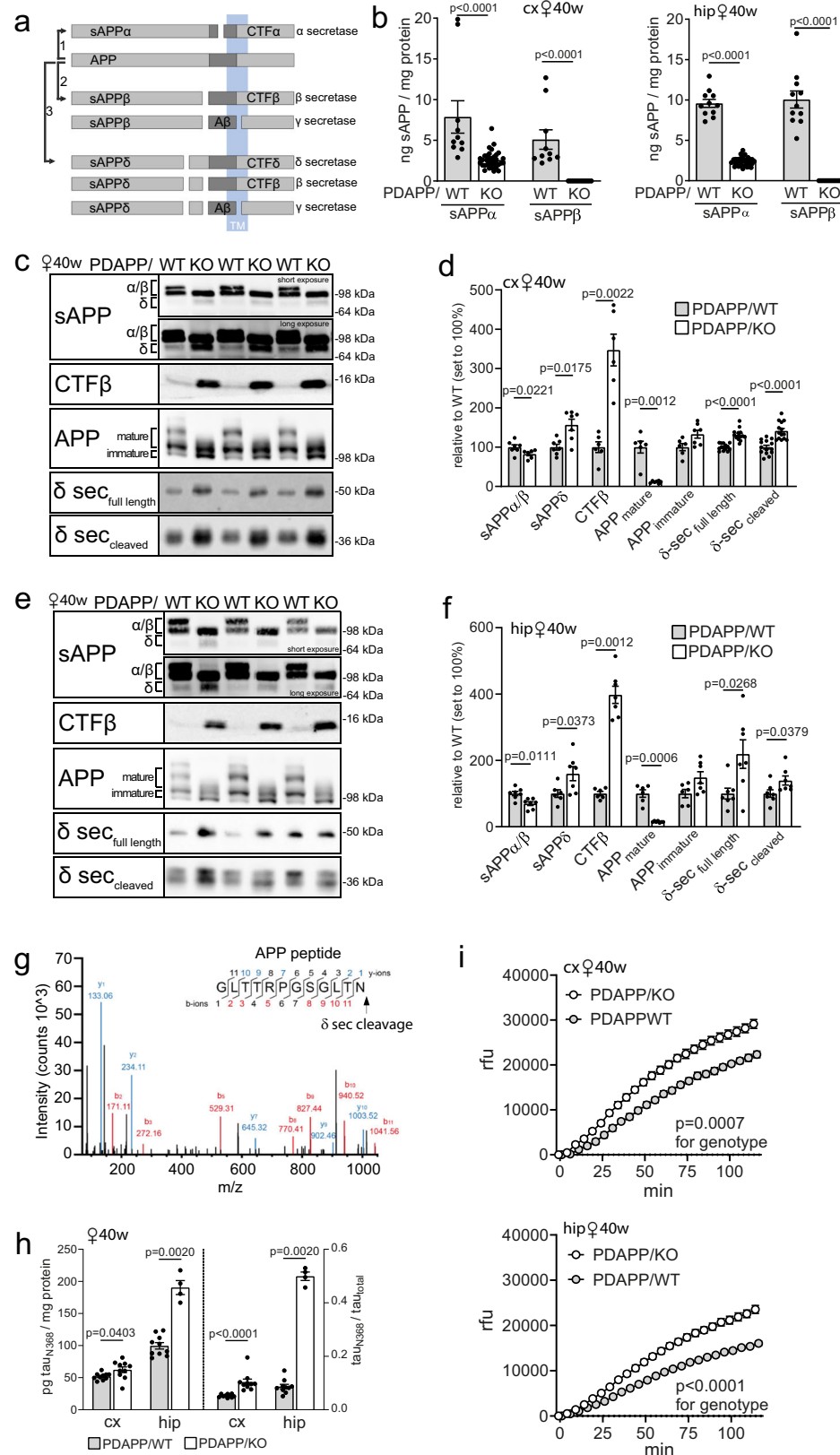

protein extracts were obtained by homogenization of brain tissues in lysis buffer directly.

For Western blotting, protein extracts from mouse brain tissues or primary astrocyte cultures were subjected to SDS-PAGE in Tris-glycine gels, followed by transfer to nitrocellulose membranes. For immunodetection, the following primary antibodies (diluted in 20 mM Tris, 150 mM NaCl, 0.1% Tween-20, 5% BSA) were used: APP antiserum (1227, in-house), δ secretase (Legumain #93627, Cell Signaling Technology), SORCS2 (AF4237, R&D Systems), IBA1 (ab5076, Abcam), GFAP (G3893, Sigma Aldrich; #20334, DAKO Agilent), cleaved caspase-3 (#9664, Cell Signaling Technology), cleaved PARP (Asp214, #9541, Cell Signaling Technology), tau (HT7, Invitrogen #MN1000), ptau (AT8,

**Fig. 9 | Enhanced δ secretase activity in the SORCS2-deficient mouse brain. a** Schematic presentation of APP processing pathways and the resulting cleavage products. (1) non-amyloidogenic processing initiated by α secretase, (2) amyloidogenic processing by β and γ secretases, (3) alternative amyloidogenic processing by sequential cleavage from δ, β, and γ secretases. CTFδ and sAPPδ are cleavage products unique to δ secretase action. **b** Levels of soluble (s)APPα and sAPPβ in cortex (cx) and hippocampus (hip) of 40 weeks old female PDAPP/WT and PDAPP/KO animals as determined by ELISA. Data are given as mean ± SEM from $n = 10$ (cx, WT), $n = 11$ (hip, WT), $n = 12$ (cx, sAPPβ KO), $n = 14$ (hip, sAPPα KO), $n = 35$ (cx and hip, sAPPα KO) animals per group (two-sided unpaired Mann-Whitney U test for cx, two-sided unpaired Student's $t$-test for hip). Representative Western blot (**c**), and densitometric scanning of replicate blots (**d**), quantifying levels of APP, CTFβ, sAPP species, as well as full-length and cleaved forms of δ secretase in the cx lysates of 40 weeks old PDAPP/WT and PDAPP/KO females. Immunoreactive bands representing mature and immature forms of APP as well as APPα/β and sAPPδ are marked. Data are expressed as relative to WT (set to 100%) and given as mean ± SEM from $n = 6$ (WT CTFβ, APP mature, APP immature; KO CTFβ), $n = 7$ (KO APP mature, APP immature; WT and KO sAPPα/β, WT and KO sAPPδ), $n = 14$ (δ secretase WT and KO) animals per group (two-sided unpaired Mann-Whitney U test for APP,

CTFβ, and sAPP; two-sided unpaired Student's $t$-test for δ secretase). **e, f** Experiment as in (**c, d**), but testing hip extracts of 40 weeks old PDAPP females. Data are expressed as relative to WT (set to 100%) and given as mean ± SEM from $n = 6$ (WT CTFβ, APP mature, APP immature, sAPPδ), $n = 7$ (all other groups) animals per group (two-sided unpaired Student's $t$-test for sAPPδ and two-sided unpaired Mann-Whitney U test for other comparisons). **g** Peptide sequence of proteolytic fragment produced from recombinant APP, following incubation with PDAPP/KO brain extracts, verifies APP cleavage after N585[43]. **h** Levels of truncated tau (tau$_{N368}$) and ratio of tau$_{N368}$/tau$_{total}$, in cortex and hippocampus of 40 weeks old female mice as determined by ELISA. Data are given as mean ± SEM from $n = 4$ (KO, hip tau and hip tau$_{N368}$/tau$_{total}$), $n = 10$ (all other groups) animals per group (two-sided unpaired Mann-Whitney U test). **i** Activity of δ secretase in cx and hip lysates of 40 weeks old PDAPP/KO and PDAPP/WT females, as documented by activity assays using a fluorogenic substrate (see methods for details). Data are given as mean ± SEM from $n = 6$ (WT), $n = 7$ (KO) animals per group (repeated measures two-way ANOVA to test significance for genotypes). $p$-values for all statistically significant differences are indicated on the graphs. Source data are provided in the Source Data file.

Invitrogen #MN1020; AT180; Invitrogen # MN1040), GAPDH (Gene-Tex, #GTX627408) or beta actin (Abcam #ab8227).

For ELISA, soluble protein fractions of brain extracts were used to measure various analytes. APP processing products were determined by multiplex assays (V-PLEX Plus Aβ Peptide Panel 1 (4G8), #K15199G; sAPPα/sAPPβ Kit, #K15120E; Meso Scale Discovery). Levels of IL-1β, TNFα, IL-12p70, IL-6, IP10/CXCL10, IL-33, MIP1α/CCL3, IL-2, IL-5, IL-16, IL-10, IL-4, and IL-22 were determined by custom-made multiplex assays using the Meso Scale Discovery protocol. TGF-β2 /TGF-β3 levels were determined using the U-PLEX TGF-β Combo multiplex assay (#K15242K, Meso Scale Discovery). Phospho and total tau levels were measured using Kit #K15121D (Meso Scale Discovery). Truncated tau levels were determined using Kit #ab315059 (Abcam). PikoKine ELISA was used to quantify YKL40/Chi3l1 and IL-33 (Boster Biological Technology #EK0975, #EK0930) according to the manufacturer's protocols.

### Quantitative RT-PCR

Total RNA was extracted from brain tissues using the RNeasy Plus Micro Kit (Qiagen # 74034) combined with the RNase-free DNase Set (Qiagen # 79254) or from primary astrocyte cells using the Direct-zol RNA Miniprep Kit (Zymo Research #R2052) according to the manufacturers' instructions. The RNA was converted to cDNA (Applied Biosystems # 4387406; 4368814) and gene expression analysis (Applied Biosystems # 4369016; 4366072, 4385612) was performed using the following probes: *Sorcs2* (Mm00473050_m1 and Mm01217942_m1), *Iba1/Aif1* (Mm00479862_g1), *Tmem119* (Mm00525305_m1), *Cdh5* (Mm00486938_m1), *Vwf* (Mm00550379_m1), *Mbp* (Mm01266402_m1), *Mog* (Mm01273867_m1), *Gfap* (Mm01253033_m1), *Aldh1l1* (Mm03048949_m1), *NeuN/Rbfox3* (Mm01248771_m1), *Baf53b/Actl6b* (Mm00504274_m1), *Cd32* (Mm 00438875_m1), *Cd40* (Mm 00441891_m1), *Cd163* (Mm 00474091_m1), *Cd206* (Mm 01329359_m1), *Lgmn* (Mm01325350_m1), *Rn18s* (Mm03928990_g1), *LGMN* (Hs00271599_m1), *GAPDH* (Hs02758991_g1) (Thermo Fisher Scientific). Primers used for amplification for murine *Lgmn* in the co-culture experiments were: Fw, ACTGGTACAGCGTCAACTGG; Rev, GTGTGGGACTTGACCAGGTG.

### Immunohistochemistry

Mice were perfused and fixed with 4% paraformaldehyde in PBS. Brains were carefully dissected and postfixed for additional 24 h before treatment in 30% sucrose/PBS for several days at 4 °C. Free-floating 40–50 µm sections were processed by 10 min antigen retrieval in 10 mM citric acid/TBA (20 mM Tris, 150 mM NaCl pH 6.0) at 80 °C, followed by 10 min permeabilization in PBS/0.05% Tween-20 with 0.3% Triton X-100, and blocking with M.O.M. (Vector Labs

#MKB-2213) for 60 min. In case of GFAP and IBA1 staining, antigen retrieval and permeabilization were not performed. Next, free-floating sections were blocked for 1 h in 1% horse serum in PBS. For immunodetection, the sections were incubated overnight at 4 °C with primary antibodies diluted in incubation buffer (PBS with 1% bovine serum albumin, 1% normal donkey serum, 0.3% Triton X-100). Then, sections were washed in PBS and treated for 2 h at room temperature with fluorochrome-conjugated secondary antibodies (Alexa, Invitrogen). After washing in DAPI, stained sections were mounted in DAKO Fluorescence Mounting Medium (F4680, Sigma Aldrich). The following primary antibodies were used: tau (HT7, Invitrogen #MN1000; cleaved tau (tauC3), Invitrogen #AHB0061), ACSA2 (Miltenyi Biotec #130-116-245), GFAP-Cy3 (Sigma #C9205), 6E10 (Bio Legend #803014), MAP2 (Cell Signaling #8707), CD31 (Novus Biologicals #NB600-1475) as well as IBA1 (Wako #019-198741). Quantifications of signal intensities and area occupied by amyloid plaques were performed in ImageJ. Biotinylated phospho-tau antibodies AT8, Ser202, and Thr205 (Invitrogen #MN1020B) were used to visualize tau. Quantification was performed using the Colour Deconvolution with H DAB function in ImageJ. For co-immunodetection of SORCS2 with NeuN or S100β, tissue sections were pre-treated for 30 min at room temperature in PBS containing 0.1% SDS and 50 mM DTT, followed by permeabilization in PBS containing 0.02% saponin for 10 min. Next, sections were blocked for 1 h at room temperature in blocking solution (PBS, 0.1% saponin, 5% BSA, and 10% normal donkey serum (Sigma-Aldrich, D9663). Primary antibodies were applied overnight at 4 °C, either in ready-to-use S100β antibody solution or in PBS, 0.1% saponin buffer (for NeuN). Primary antibodies used were rabbit anti-S100β (Agilent, GA504, ready-to-use), sheep anti-SORCS2 (R&D Systems, AF-4237, 1:100), and biotin-conjugated mouse anti-NeuN (Sigma, MAB377B). Next, sections were washed and then incubated with Alexa Fluor 555 donkey anti-rabbit IgG (1:250), Alexa Fluor 488 donkey anti-sheep IgG (1:250), and Streptavidin Alexa Fluor 555 Conjugate (1:200) in PBS,1% donkey serum for 1 h at room temperature. Sections were washed again with PBS, including DAPI (Sigma-Aldrich, D9542, 1 µg/ml) during the final wash, air-dried, and mounted in DAKO fluorescent mounting medium (S3032) for imaging.

For visualization of amyloid deposits, mouse brains were fixed in 4% paraformaldehyde/PBS for 24 h. After dehydration, tissues were embedded in paraffin and sectioned on Super Frost Plus glass slides at 5 µm. Sections were deparaffinized and rehydrated with a series of Roti-Histol and ethanol, and stained with 1% aqueous Thioflavin-S solution (Sigma #T1892) for 8 min at room temperature. Thereafter, the sections were washed twice in 80% and once in 95% ethanol, followed by three washes in distilled water.

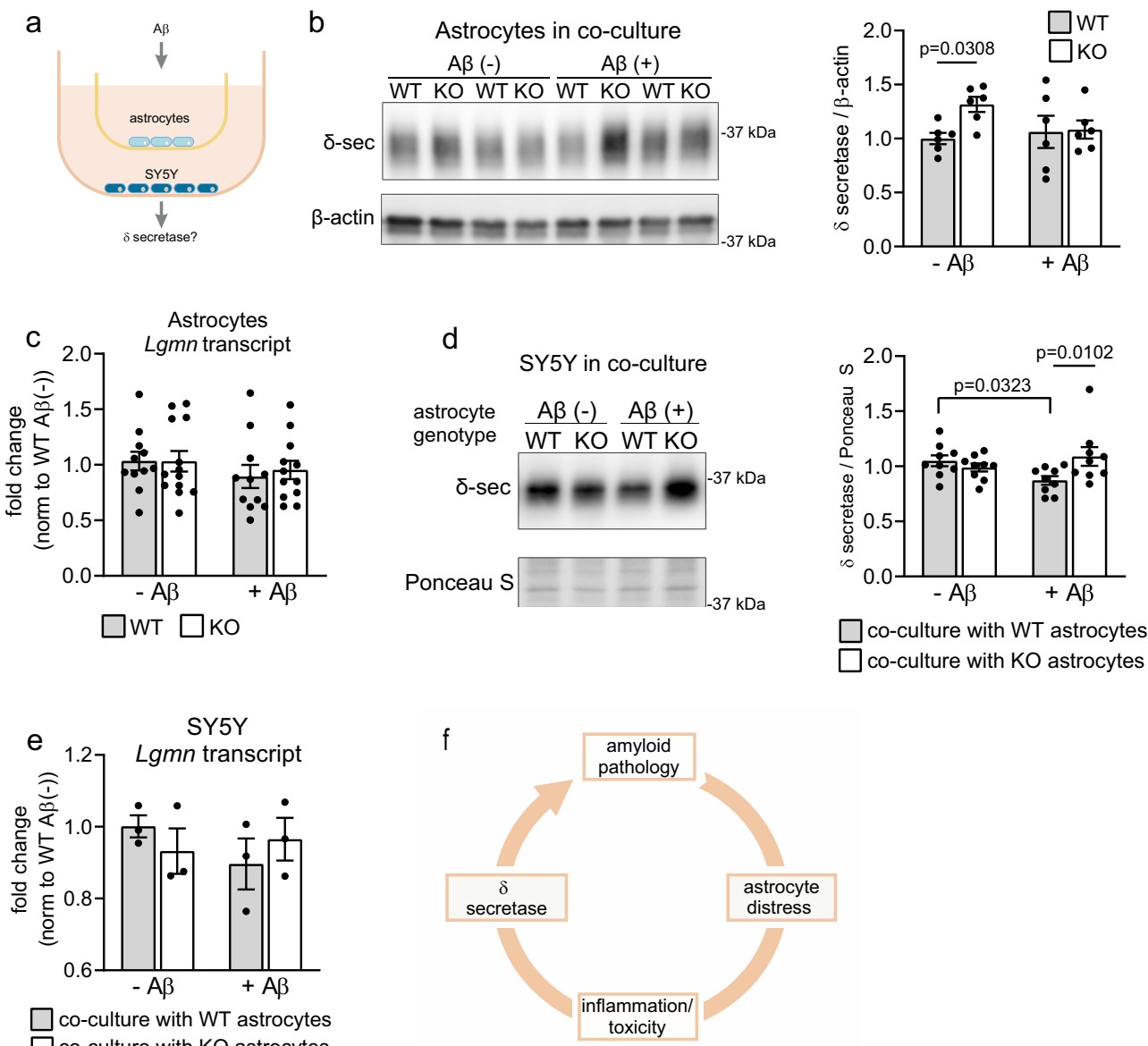

**Fig. 10 | Exposure of SORCS2-deficient astrocytes to Aβ induces expression of δ secretase in neuroblastoma cells in trans. a** Experimental setup for co-culture model of primary astrocytes and SH-SY5Y cells, exposed to Aβ stress. SH-SY5Y cells (bottom well) were co-cultured with primary murine WT or KO astrocytes (in insets). The co-cultures were incubated in medium containing Aβ or control medium. The corresponding response in δ secretase transcript and protein levels was determined in both cell types. **b** Representative Western blot analyses, and densitometric quantification of replicate blots thereof, document levels of δ secretase in the lysates obtained from WT and KO astrocytes treated as described in (**a**). Detection of β-actin served as loading control. Data are expressed as relative to WT Aβ- (set to 1) and given as mean ± SEM from $n = 6$ biological replicates. Two-Way ANOVA with Uncorrected Fisher´s LSD. **c** Levels of *Lgmn* transcripts as determined by qRT-PCR in primary WT and KO astrocytes treated with control (-) or Aβ-conditioned media (+) for 24 h. Data are expressed as relative expression levels (set

to 1 for WT Aβ-) and given as mean ± SEM from $n = 11$ (WT), $n = 12$ (KO Aβ+), $n = 13$ (KO Aβ-) biological replicates per group. **d** Representative Western blot analyses, and densitometric quantification of replicate blots thereof, document levels of δ secretase in the lysates obtained from SH-SY5Y cells treated as described in (**a**). Ponceau S staining was used as a loading control. Data are expressed as relative to WT Aβ- (set to 1) and given as mean ± SEM from $n = 9$ biological replicates. Two-Way ANOVA with Uncorrected Fisher´s LSD. **e** Experiment as in (**c**), quantifying levels of *Lgmn* transcripts in SY5Y cells treated as described in (**a**). Data are expressed as relative expression levels (set to 1 for WT-Aβ) and given as mean ± SEM from $n = 3$ biological replicates per group. This experiment was performed once. **f** Model of pathological cascade induced by Aβ stress imposed on astrocytes (see discussion section for details). *p*-values for all statistically significant differences are indicated on the graphs. Source data are provided in the Source Data file.

**Primary astrocyte cultures**

To obtain adult primary astrocytes for Aβ uptake assays, cortices of 40 weeks old PDAPP female mice were dissected and prepared as described for FACS sorting above. Cell pellets were kept on ice during preparation of the isotonic gradients by diluting Percoll (GE Healthcare #17-0891-01) in 10x PBS to a stock solution of 1.12 g/ml in 1x PBS (100% SIP). The stock solution was further diluted in PBS to obtain 70% SIP,

50% SIP, and 35% SIP. The cell pellets were resuspended in 4 ml of 35% SIP and loaded onto a gradient of 4 ml of 50% and 2 ml of 70% SIP in a 15 ml Falcon tube at room temperature. The prepared tubes were centrifuged in a swingout bucket at 2000 x *g* with acceleration 3 and break 1 for 20 min. After centrifugation, the top myelin layer was removed and cells from the interface between 35–50% SIP were collected in a new 15 ml Falcon tube. The isolated astrocytes were washed and

resuspended in DMEM/F12 medium (Gibco #11320033) containing 10% fetal bovine serum (Life Technologies, #10270106) and 1% penicillin/ streptomycin (Life Technologies, #15140-122), and seeded in 24-well culture dishes pre-coated with 0.1 mg/ml poly-D-lysine hydrobromide (Sigma #P1274). For Aβ uptake, astrocytes were washed with PBS and incubated for 2 h with 200 μl growth medium containing 1 μM Aβ40 HiLyte-488 (Ana Spec #AS-60491-01). Next, the cells were washed three times and imaged for fluorescence signals from HiLyte-488. Thereafter, cells were trypsinized (Gibco, 0.25% trypsin/EDTA, #25200056) and washed in PBS. The suspensions were centrifuged at 2500 x g for 5 min, and the cell pellets lysed in lysis buffer (20 mM Tris-HCl, 10 mM EDTA, 1% NP40, 1% Triton X-100, pH 7.4) for 1 h on ice. The intracellular Aβ content was quantified by ELISA (Meso Scale Discovery).

To obtain primary astrocytes from neonates, brain tissues were dissected from mouse pups (both sexes; sex was not determined) with olfactory bulbs, cerebellum, and meninges removed. Tissues were washed three times at room temperature with HBSS (Gibco #14175-095) leaving approximately 1.5 ml of buffer after the last wash. Two hundred μl of solution containing 100 mg/ml trypsin (Sigma-Aldrich #T8003) and 5 mg/ml DNase I (Sigma-Aldrich #DN25) was then added and incubated for 2 min at room temperature with brief shaking. The process was stopped by adding 5 ml of Dulbecco's modified Eagle's medium (4500 mg/l glucose, 1 mM sodium pyruvate) supplemented with 4 mM glutamine (Gibco #31966-021), 10% (vol/vol) FBS (Gibco #10500064), and 1% of penicillin/ streptomycin (Sigma #P4333). After removing the medium until approximately 1.5 ml remained, 200 μl of 5 mg/ml DNase I solution was added and the tissue was disrupted by pipetting with a Pasteur pipette. Subsequently, 10 ml of fresh medium were added and the cell suspensions centrifuged at 120 x g for 10 min at room temperature. Thereafter, the cell pellet was resuspended in fresh medium (1 ml per 2 brains) and plated in a T75 flask (Sarstedt #83.3911.302) pre-coated with poly-L-lysine hydrobromide (0.1 mg/ml final concentration, Sigma #P1274). At DIV2, flasks were washed three times with 1x PBS (Gibco #14200) and 15 ml of fresh cell medium were added. At DIV10, flasks were shaken on a horizontal shaker at 80 rpm and 37 °C for 60 min to remove microglia population. Fresh medium was added to the remaining astrocytes. At DIV15, flasks were shaken overnight at 180 rpm, washed three times with 1X PBS, trypsinized, and cells plated on Petri dishes. At DIV20, cells were harvested and stored in liquid nitrogen. For each experiment, WT and KO astrocytes were thawed, plated, and reseeded after 3–5 days.

## Co-culture model of astrocytes and neuroblastoma cells

Astrocytes and SH-SY5Y human neuroblastoma cells (CRL-2266, ATCC) were initially seeded separately. Astrocytes were plated on Falcon cell culture inserts (Corning #353090) at a density of $4 \times 10^5$ cells per insert in DMEM/HAMS F12 (Life Technologies #21331046) medium supplemented with 10% FBS (Life Technologies #10270106), GlutaMAX (Life Technologies #35050-038), NEAA (Life Technologies #11140-035), and penicillin/streptomycin (Sigma #P4333). Neuroblastoma cells were seeded in 6-well plates at $2 \times 10^5$ cells per well. After 24 h, the astrocyte cell inserts were transferred into the wells containing neuroblastoma cells, and the medium in both cell compartments was replaced with media 24 h-conditioned by parental SH-SY5Y cells (Aβ-) or a subclone stably overexpressing human APP[21] (Aβ + ). Following an additional 24 h incubation, cells were harvested separately for Western blot and qPCR analyses.

## Mass spectrometry

Cyanogen bromide (CNBr)-activated Sepharose beads (30 mg) were swollen in 1 mM HCl for 10 min, washed 5x with 1 mM HCl and 3x with coupling buffer (100 mM sodium bicarbonate, 500 mM sodium chloride, pH 8.3). Purified recombinant human APP751 (10 mg; Bio Legend #842601) was incubated with the beads in coupling buffer over night at 4 °C. Thereafter, the beads were washed with coupling buffer,

blocked with 100 mM Tris-HCl, pH 8, for two hours at 4 °C and washed again 5x with washing buffer 1 (100 mM sodium acetate, 500 mM sodium chloride, pH 4.0) 3x with washing buffer 2 (100 mM Tris-HCl, 500 mM sodium chloride, pH 8.0), and 3x with δ secretase assay buffer (20 mM citric acid, 60 mM Na2HPO4, 1 mM DTT, 1 mM EDTA, 0.1% CHAPS, 0.5% Triton X, pH 4.5). Cortex from PDAPP/KO animal was extracted in 1 ml assay buffer for 1 h on ice. Protein lysate was obtained after centrifugation (5 min; 20,000 x g at 4 °C) and 50 μg protein was loaded to the beads for 2 h at 37 °C with rotating. The beads were then washed several times with assay buffer and PBS.

Coupled APP was reduced by treatment with dithiothreitol and alkylated with iodoacetamide before in- solution trypsin digestion overnight. Thereafter, the peptides were purified with Pierce TM C18 Spin Tips (Thermo Scientific) and vacuum-dried before MS analysis. Peptide samples were trapped and separated by liquid chromatography (Easy-nLC 1200, Thermo Scientific) using precolumn (Acclaim PepMap 100, 75 μm × 2 cm, Nanoviper, Thermo Scientific) and analytical column (EASY-Spray column, PepMap RSLC C18, 2 μm, 100 Å, 75 μm × 25 cm) in a 10 min gradient of 4–40% acetonitrile in 0.1% formic acid, coupled to the mass spectrometer Q-Exactive HF-X Hybrid Quadrupole Orbitrap (Thermo Scientific, Bremen). Precursor ion mass spectra (MS1) were acquired at 60,000 resolution, and the scan range was between 372 to 1800 m/z. MS2 analysis was performed in a data-dependent mode, where the most intense doubly or multiply charged precursors were fragmented. MS2 resolution was set at 15,000. Unassigned and +1 charge state was excluded from fragmentation, and a dynamic exclusion of 15 s was used. To identify cleavage sites on human APP, the Thermo Proteome Discoverer 3.0.0.757 with SEQUEST HT algorithm was used to search and match MS/MS spectra to a human sequence database (Homo Sapiens proteome with 22,763 reviewed sequences downloaded from Uniprot database, https://www.uniprot.org/, May 2020), including delta-cleaved GLTTRPGSGLTN as an additional peptide.

## Data acquisition and analysis

Confocal images were acquired using Zeiss LSM 700 and Zen 2012 software. Western blot signals were captured with a Bio-Rad ChemiDoc Imager and Image Lab Touch software (version 2.4.0.03). Lysotracker signal imaging was performed with the Opera Phenix microscope (Revvity). Presto Blue fluorescence measurements were obtained using the VICTOR3 1420 Multilabel Counter (Perkin Elmer) with Victor2030_v4_XP software. Flow cytometry data were collected using Aria Fusion Software (BD). ELISA absorbance measurements were performed using the Mesoscale Discovery platform with Discovery Workbench 4.0.12 and the Tecan Spark 10 M plate reader with Spark Control V1.2.20. qPCR data were collected using the StepOne-Plus Real-Time PCR System (Applied Biosystems) with StepOne software version 2.2.2. Western blot analysis was conducted using Image Studio Lite (version 5.2.5) and ImageJ (version 1.52p). Flow cytometry data were analyzed with FlowJo 10. Microscopy images were analyzed with ImageJ (version 1.52p), except for lysotracker signal analysis, which was done using Harmony 4.9 software (Revvity). qPCR data analysis was done with StepOne software version 2.2.2. Mass spectrometry data were analyzed with Thermo Proteome Discoverer 3.0.0.757. Statistical analysis was done in GraphPad Prism 8 or 10.

## Statistical information

For all experiments, an indicated number n is the number of mice per group used in an experiment. Each mouse and each preparation of primary astrocytes represents a statistically independent experimental unit, which was treated accordingly as an independent value in the statistical analysis. Unless stated otherwise in the figure legend, each experiment was repeated at least twice with similar results. Statistical analyses were performed using GraphPad Prism 8 or 10 Software. Data are presented as mean ± standard error of the mean (SEM), mean ±

standard deviation (SD) or geometric mean ± confidence interval (CI) as indicated in the respective figure legends. To compare two groups, unpaired two-sided Student's *t* test or unpaired two-sided Mann–Whitney U test was used, depending on normal distribution of data. For experiments with more than two parameters, an ordinary or repeated-measures Two-way ANOVA with Tukey's or Holm-Sidak's multiple comparison test was applied. Before selecting a test, a preliminary analysis for normal distribution of the data was performed. The exact test used is indicated in the figure legends.

### Reporting summary

Further information on research design is available in the Nature Portfolio Reporting Summary linked to this article.

## Data availability

Source data are provided with this paper. All numerical source data and uncropped images of immunoblots are published alongside the paper as Source Data file. The mass spectrometry data have been deposited to the ProteomeXchange Consortium via the PRIDE partner repository with the dataset identifier PXD067552. Source data are provided with this paper.

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

## Acknowledgements

The authors are grateful to K. Kampf, T. Pasternack, and R. Vogel for expert technical assistance. We are indebted to the Netherlands Brain Bank and the MRC London Brain Bank for Neurodegenerative Diseases for providing brain tissue samples. This work was supported by the National Science Center OPUS program (2020/37/B/NZ3/00761, ARM), Research University Program at the University of Warsaw (I.3.4 Action of the Excellence Initiative, ARM), Alzheimer Forschung Initiative (#23001 R, TEW), and Novo Nordisk Foundation (NNF18OC0033928, TEW).

## Author contributions

V.S., A.R.M., and T.E.W. conceptualized the study and wrote the manu-script. V.S., E.Z., TO, JPG, EZ-P, J.C., J.P., and A.R.M. performed

experiments and evaluated data. B.L.H. contributed essential mouse lines, conceptualized experiments, and evaluated data.

## Funding

## Competing interests

The authors declare no competing interests.
