## [Transparent Peer Review file · Nature Communications]

Astrocytes distress triggers brain pathology through induction of δ secretase in a murine model of Alzheimer's disease

Corresponding Author: Professor Thomas Willnow

Version 0:

Reviewer comments:

Reviewer #1

(Remarks to the Author)

In this manuscript, Schmidt et al. investigate the role of the protective stress-response receptor SORCS2 in mediating the response of astrocytes to β amyloid, and serves a role in reducing stress and pathology of astrocytes in an APP model (PDGF β promoter driving APP with the Indiana mutation). SORCS2 also plays a role in suppressing delta-secretase which promotes amyloidogenic and tau-related proteolytic processing. It is postulated that the lack of SORCS2 leads to progressive glial pathology associated with neuroinflammation, β amyloid deposition, tauopathy, and astrocytic pathology/death. The study uses a combination of in vitro and in vivo methods to study the expression and function of SORCS2 in glial reactivity, neuroinflammation, β amyloid processing/deposition, and tauopathy. Both full KO and conditional (astrocytic and neuronal) KOs are used to test cell-type contributions of SORCS2 to these pathological events.

The strength of the paper lies in the robustness of some of the phenotypes observed in the SORCS2 KO (i.e. accumulation of A β 40/42 in the complete knock-out, promotion of hyperphosphorylation of tau (in some contexts), promotion of gliosis, and activation of delta secretase activity and generation of CTF β stubs). There is also evidence of some astrocyte-specific contributions to these phenotypes. However, while the study makes several interesting observations, including the exacerbation of AD-related pathologies upon loss of SORCS2, there are a number of inconsistencies in the results and gaps in the analyses that make it difficult to appreciate a clear role of SORCS2 in AD and AD-related pathology.

Major Comments:

1. The background information establishing the role of SORCS2 in disease should be better described at the beginning of the manuscript to explain SORCS2 in neurodegenerative disease (including as an AD risk gene) and its function in cellular stress and neuronal and glial cell health.
2. Astrocytes upregulate SORCS2 in vitro upon exposure to A β conditioned media (Figure 1). However, the majority of the analysis involves either complete loss of SORCS2 in all cells or specifically loss of SORCS2 in astrocytes. An important question is whether SORCS2 expression is up or down regulated locally or globally (and in which cells), in AD and in AD models. This is needed to better understand the relevancy of the loss-of-function models for AD and progressive increases in AD-related pathology.
3. The result showing accumulation of A β plaques and hyperphosphorylation of tau presented in Figures 2 and 3 need to be quantified. When do these = plaques and tau alterations appear in PD/APPKO mice and is the time course for the development and progression of these pathological events different for cortex and hippocampus? Also, which cells show altered tau? From Figure 3, it appears that HT7 and cleaved tau signals are enriched around blood vessels. Double-labeling for neuronal, glial, and endothelial markers would help interpret which cells are showing tau pathology.
4. The reported sex-specific effects are hard to follow throughout the manuscript. Sometimes it appears that the analysis is specific to females or males. These results are interlaced with analysis of both sexes studied together. For example, lines 72-74 and related to Figure 2c, is this analysis of PDAPP/KO female mice or both male and females? The previous sentence suggests this is analysis of female mice only. However, the subsequent sentence (40 week analysis) suggests

both males and females were analyzed together. More clear presentation of the sex-specific analysis and differences are needed as well as modification of the figures to indicate if the analysis is on female, male, or both sexes of mice.

5. FACS sorting with ACSA2 indicates a loss of astrocytes in PDAPP/KO mice (Ext. data 4). However, analysis of PDAPP/KO mice suggests a reduction of ACSA2 and S100 β protein (Figure 2g). Thus it remains unclear if the suggested reduction in astrocytes is due to lower ACSA2 FACS sorting ability or a reduction in cells. Quantification of SOX9+ astrocytes in PDAPP/KO vs. PDAPP/WT brains would help resolve this.

6. The astrocyte specific knock-out does not show a robust effect in the cortex even though the loss of SORCS2 transcripts is just as efficient in cortex as in hippocampus where the phenotype is present. The loss of effect in cortex is problematic as the efficiency of the SORCS2 seems comparable in cortex and hippocampus (Ex. data figure 5) but the effects on A β 40 and A β 42 are lost (compare Figure 6b, left panel, with Figure 2 analysis of cortex).

7. The neuronal-specific knock-out appears to show some non-cell autonomous effects on astrocytes. The Western blot analysis shown in Ex data 5 suggest a large loss of SORCS2 protein with the neuron-specific KO. Could it be that neurons have lower SORCS2 transcript levels but higher protein levels than astrocytes? This might explain the loss of impact in cortex in the astrocyte-specific KO. This can be analyzed in more detail.

8. If the mechanism leading to altered delta secretase relies on astrocytic-expressed SORCS2, why use the full knock-out to study delta secretase? This was not well justified.

9. How does delta secretase contribute to the phenotypes directly? Is delta cleavage of APP detected in PDAPP mice? Is it possible that the enhanced delta secretase activity is unmasked following a complete loss of SORCS2 that normally does not happen in the context of AD in humans and AD models? This gets back to points 1 and 2 above.

Minor points

1. Add labeling of age and sex of mice to graphs.

2. Line 70-71, Figures 2b and 2b, regarding the statement, "PDAPP/KO females showed an age-dependent increase in brain cortex levels of soluble A β 40 and A β 42 compared to PDAPP/WT, starting from 12 weeks of age (Fig. 2a-b)." This statement is a bit misleading in that it suggests that PDAPP/KO mice have significant and consistent changes in both A β 40 and A β 42 from 12 weeks onward. However, PDAPP/KO mice do not show significant changes in A β 40 until 25 weeks. Also, A β 42 is not significantly different at the 16 and 19 week timepoints in PDAPP/KO according to the analysis.

Reviewer #2

(Remarks to the Author)

The manuscript by Schmidt et al. shows that astrocyte distress induced by A β stress promote A β pathology through d secretase (AEP). Specifically, they first found that Sorcs2 KO astrocytes are more vulnerable to A β stress both in vitro and in vivo. Therefore, they further use Sorcs2 deletion as a model to study heightened astrocyte stress in AD. SORCS2 deficiency induces A β pathology and tau phosphorylation in AD mice, and astrocytes are identified as the most impacted cell type. The mechanism is explained that Sorcs2 deletion provokes microglia activation and pro-inflammatory response in AD mouse. Therefore, this leads to the activation of d secretase (AEP) to overproduce A β .

The authors need to address some major unanswered questions and concerns prior to publication:

1. The authors indeed see the increased tau fragments (D421/D422) as tau seeds in PDAPP/KO brain. Tau N368 fragment is essential for tau aggregation produced by d secretase, thereby tau N368 should be detected as major tau seeds in Figure 3 and Figure 6.

2. Global PDAPP/KO mice show increased tau phosphorylation and fragmentation, however, in the PDAPP/asKO mice, the tau pathology is not shown. As the importance of tau pathology in AD, tau phosphorylation and fragmentation should also be detected for PDAPP/asKO mice.

3. A β antibody staining (eg. 6E10) or X34 staining are well-established methods to evaluate A β pathology. At least one of the methods should be used for PDAPP/KO mice in Figure 2 and PDAPP/asKO mice in Figure 6.

4. In figure 7, antibody specific for APP N585 should be used to detect the cleavage of APP by AEP.

5. The author claimed the overproduction of A β is caused by activated d secretase (AEP) in PDAPP/KO mice. Although AEP activation is observed in PDAPP/KO mice, this could only be proved when inhibition of AEP (by genomic deletion or inhibitor) blocks the A β overproduced.

Other minor comments:

1. Many figures fails to show which groups the p values are referring to. Besides, the p value for Figure 1F left panel is missing.

2. IF images from Figure 1C should be quantificated.

3. Enternal control (eg. GAPDH) should be introduced for Figure 1E, Figure 5E and Figure 6D.

Reviewer #3

(Remarks to the Author)

The manuscript by Schmidt et al provides an interesting group of studies implicating astrocytic SORCS2 signaling in Alzheimer's disease. Evidence mostly comes from global and astrocyte specific KO of SORCS2 in PD/APP mice. The results show that SORCS2 KO increases GFAP levels, neuroinflammation, and elevated Abeta and phospho-tau levels. The authors suggest that the effects of SORCS2 KO are mediated by the activation of delta secretase.

Many of the SORCS2 KO data are compelling, but there are some major conceptual issues that limit the impact of the results, which are outlined below along with a few technical questions.

1. Evidence showing higher pathology loads in astrocyte-specific SORCS2 KO in mice is compelling, but the actual significance or relevance of this KO to AD is not clear. Does SORCS2 expression change in astrocytes during AD? Human data would greatly strengthen the validity of this pathway in the context of disease. Assessing changes in SORCS2 levels in astrocytes as a function of aging in the PD/APP model would be important to include as well. If SORCS2 changes in astrocytes were important to the pathophysiology of AD, one would expect these changes to occur early in the course of disease.
2. Along the same lines, the title of the manuscript should be changed to reflect that this was a mouse study, and not a study on actual Alzheimer's disease.
3. Another conceptual issue is that SORCS2 KO appears to kill astrocytes, but there doesn't seem to be much astrocyte loss in AD (again, this gets at whether the results are relevant to human disease). As astrocytes are protective cells by nature, it seems likely that any manipulation that kills astrocytes would also exacerbate neuropathology?
4. Technical question: Does SORCS2 KO also kill astrocytes at 20 weeks of age (maybe this explains the opposing results of the KO in 20 week old vs 40 week old mice)?
5. Technical question: The FACS data suggest SORCS2 transcript expression is higher in astrocytes than other neural cell types, but is SORCS2 protein expression enriched in astrocytes? The lack of immunohistochemical/fluorescent evidence is a major weakness of the study.
6. Technical question: in the FACS sorting protocol, it appears that neurons are identified by process of elimination (negative for CD45, C49a, O4, and ACSA2). Neurons can be tricky to sort from intact brain tissue. Often many of the neuronal processes are lost in these procedures. This could introduce a bias, as many mRNA species are found in neuronal processes. Is it possible that the percentage of neurons expressing Sorcs2 (shown in Figures 4B and C) is underestimated?
7. Technical issue: the delta secretase results are interesting, but whether these changes occur specifically in reactive astrocytes is highly speculative.

Version 1:

Reviewer comments:

Reviewer #1

(Remarks to the Author)

The authors have sufficiently addressed my previous comments through additional analysis and more thorough explanation.

Reviewer #2

(Remarks to the Author)

The revised manuscript has successfully addressed my previous concerns. The paper is now merits of publication!

Reviewer #3

(Remarks to the Author)

See previous review comments. The authors have satisfactorily addressed all of my concerns.

REPLY TO REVIEWERS' COMMENTS

Reviewer #1 (Remarks to the Author):

In this manuscript, Schmidt et al. investigate the role of the protective stress-response receptor SORCS2 in mediating the response of astrocytes to β amyloid, and serves a role in reducing stress and pathology of astrocytes in an APP model (PDGF β promoter driving APP with the Indiana mutation). SORCS2 also plays a role in suppressing delta-secretase which promotes amyloidogenic and tau-related proteolytic processing. It is postulated that the lack of SORCS2 leads to progressive glial pathology associated with neuroinflammation, β amyloid deposition, tauopathy, and astrocytic pathology/death. The study uses a combination of in vitro and in vivo methods to study the expression and function of SORCS2 in glial reactivity, neuroinflammation, β amyloid processing/deposition, and tauopathy. Both full KO and conditional (astrocytic and neuronal) KOs are used to test cell-type contributions of SORCS2 to these pathological events.

The strength of the paper lies in the robustness of some of the phenotypes observed in the SORCS2 KO (i.e. accumulation of A β 40/42 in the complete knock-out, promotion of hyperphosphorylation of tau (in some contexts), promotion of gliosis, and activation of delta secretase activity and generation of CTF β stubs). There is also evidence of some astrocyte-specific contributions to these phenotypes. However, while the study makes several interesting observations, including the exacerbation of AD-related pathologies upon loss of SORCS2, there a number of inconsistencies in the results and gaps in the analyses that make it difficult to appreciate a clear role of SORCS2 in AD and AD-related pathology.

Major Comments:

1. The background information establishing the role of SORCS2 in disease should be better described at the beginning of the manuscript to explain SORCS2 in neurodegenerative disease (including as an AD risk gene) and its function in cellular stress and neuronal and glial cell health.

We apologize for insufficient description of the background information. We now have extended the introduction section to highlight prior important work that documented roles for SORCS2 in neuronal development and plasticity. We also describe earlier studies, genetically and functionally linking SORCS2 to AD, as well as its proposed role as stress response factor in neuronal and glial cell types (page 3, line 12 from below).

We trust that these revisions provide a brief yet comprehensive description of known SORCS2 functions, and that they give proper credit to key studies in the field.

2. Astrocytes upregulate SORCS2 in vitro upon exposure to A β conditioned media (Figure 1). However, the majority of the analysis involves either complete loss of SORCS2 in all cells or specifically loss of SORCS2 in astrocytes. An important question is whether SORCS2 expression is up or down regulated locally or globally (and in which cells), in AD and in AD models. This is needed to better understand the relevancy of the loss-of-function models for AD and progressive increases in AD-related pathology.

We agree that additional data on the expression of SORCS2 in the healthy versus the diseased brain provides valuable additional information on its causal role in AD. We now include new data documenting increased expression of SORCS2 in astrocytes but not neurons, sorted from AD mouse models, when compared to non-AD mice (Fig. 4b).

Importantly, enhanced receptor transcript levels were also seen in brain specimens from AD patients compared to non-AD subjects (Fig. 1j). These exciting new data from murine models and humans provide further support for the proposed role of SORCS2 as protective stress factor, uniquely upregulated in astrocytes in response to brain amyloid burden.

3. The result showing accumulation of A β plaques and hyperphosphorylation of tau presented in Figures 2 and 3 need to be quantified. When do these = plaques and tau alterations appear in PD/APPKO mice and is the time course for the development and progression of these pathological events different for cortex and hippocampus?

Quantification of the levels of hyperphosphorylated tau had already been shown in the original manuscript (Fig. 3e-f).

As suggested, we now added quantifications of the levels of A β plaques based on thioflavin S staining (Fig. 2g). These quantitative data fully substantiate the huge increase in plaque burden in aging SORCS2-deficient mice, an effect already evident from exemplary thioflavin S staining shown in the original manuscript (Fig. 2h). Plaque burden increases with age, as documented by comparative analysis of PDAPP/KO mice at 20, 30, and 40 weeks of age (Fig 2g-h).

In addition, we now corroborate a massive plaque burden in aged mutant mice by immunostainings for A β using 6E10 antibody (Fig 2i-j). These data were included in the revised manuscript at the request of reviewer 2 (point 3).

Also, which cells show altered tau? From Figure 3, it appears that HT7 and cleaved tau signals are enriched around blood vessels. Double-labeling for neuronal, glial, and endothelial markers would help interpret which cells are showing tau pathology.

As suggested, we now include co-immunostainings of tau with markers of neuronal, glial, and endothelial cell types (Fig. 3i). These data support this reviewer's conclusion of tau aggregates in SORCS2-deficient mouse brains to be primarily associated with the brain vasculature. Our new findings are in line with loss of endothelial cells in PDAPP/KO mouse brains documented by cell sorting (Fig. 4c). They also align well with earlier reports of tau aggregates associating with brain vessels, now being discussed in the revised manuscript (page 7, from top).

The corresponding text reads: "Co-immunostaining of total tau with cell type-specific markers documented more pronounced colocalization with endothelial cell marker CD31, than with markers of neurons (MAP2), astrocytes (GFAP), or microglia (IBA1) (Fig. 3i). Association of tau aggregates with the brain vasculature has also been reported in other murine models and patients with tauopathies ^{30, 31}."

4. The reported sex-specific effects are hard to follow throughout the manuscript. Sometimes it appears that the analysis is specific to females or males. These results are interlaced with analysis of both sexes studied together. For example, lines 72-74 and related to Figure 2c, is this analysis of PDAPP/KO female mice or both male and females? The previous sentence suggests this is analysis of female mice only. However, the subsequent sentence (40 week analysis) suggests both males and females were analyzed together. More clear presentation of the sex-specific analysis and differences are needed as well as modification of the figures to indicate if the analysis is on female, male, or both sexes of mice.

We regret unclear description of sexes of the mice included in the various study cohorts. We now revised all descriptions in figures, legends and result sections always indicate the sex of the animals used.

In essence, all experiments were performed in female mice, with key findings also reproduced in males. Overall, we did not observe any major sex-specific difference in SORCS2 action as stress response factor in the AD brain. Some sex-specific differences in our dataset concern technical aspects of our experimental models, namely conditional inactivation of *Sorcs2* in neurons which worked in male but not female mouse brains (Ext. data figure 5f).

The fact that phospho-tau levels increase in female (Fig. 3a-d) but not male PDAPP/KO mice (Ext. data figure 2b-e) are in perfect agreement with the sexual dimorphism observed for phosphorylation of brain tau by others. We now more clearly mention these gender-specific aspects of our findings in the result sections (page 7, line 13 from top).

5. FACS sorting with ACSA2 indicates a loss of astrocytes in PDAPP/KO mice (Ext. data 4). However, analysis of PDAPP/KO mice suggests a reduction of ACSA2 and S100 β protein (Figure 2g). Thus it remains unclear if the suggested reduction in astrocytes is due to lower ACSA2 FACS sorting ability or a reduction in cells. Quantification of SOX9⁺ astrocytes in PDAPP/KO vs. PDAPP/WT brains would help resolve this.

We trust that this reviewer refers to Fig. 4g (not 2g), documenting loss of ACSA2 immunoreactivity in brain sections from SORCS2-deficient as compared to control brains?

To confirm that the reduced number of ACSA2⁺ astrocytes sorted from mutant brains reflects a loss in astrocyte cells rather than loss of astrocytic ACSA2 expression, we repeated the FACS experiments using SOX9 as additional sorting marker (Ext. data figure 4d). These new data document loss of SOX9⁺ astrocytes in SORCS2-deficient compared to control mouse brains, unambiguously confirming astrocyte cell loss in mutant animals.

Of note, loss of various astrocytic cell populations in mutant mouse brains is also documented by FACS using sorting markers GFAP, S100b, and ALDH1L1 (Ext. data figure 4d).

6. The astrocyte specific knock-out does not show a robust effect in the cortex even though the loss of SORCS2 transcripts is just as efficient in cortex as in hippocampus where the phenotype is present. The loss of effect in cortex is problematic as the efficiency of the SORCS2 seems comparable in cortex and hippocampus (Ex. data figure 5) but the effects on A β 40 and A β 42 are lost (compare Figure 6b, left panel, with Figure 2 analysis of cortex).

We hypothesize that the differential impact of astrocytic SORCS2 deficiency on A β levels in hippocampus versus cortex in PDAPP/asKO mice reflects distinctions in astrocyte subtype composition in both brain regions as described in a recent article (www.nature.com/articles/s41467-019-14198-8). Also, differences in vulnerability reported for astrocytes in various brain regions (reviewed in PMID38521710) may explain why impaired stress response, caused by SORCS2 deficiency, is more

pronounced in hippocampal as compared to cortical brain areas. We now mention the possibility of region-specific differences in vulnerability of SORCS2-deficient astrocytes in the revised result section (page 11, line 12 from below).

While an observed increase of A β in hippocampus, but not cortex, may be considered a weakness of our conditional model, it also provides an ideal internal control to substantiate the relevance of A β as initial trigger of AD pathology. This conclusion is based on new phenotypic analyses of the astrocyte-specific KO model (PDAPP/asKO) included in the revised manuscript. In detail, astrocyte-specific loss of SORCS2 increases GFAP levels (Fig. 6d-e) and aggravates cell death (Fig. 6f) in PDAPP/asKO females. Also, levels of the early stress marker IL33 are significantly elevated in this model (Fig. 6g-h). Interestingly, increases in these features of brain stress are seen in hippocampi but not cortices of PDAPP/asKO mice, in line with A β levels being only elevated in the hippocampus.

The conclusions from these important new findings are detailed in the revised result section (page 12, line 8 from top).

7. The neuronal-specific knock-out appears to show some non-cell autonomous effects on astrocytes. The Western blot analysis shown in Ex data 5 suggest a large loss of SORCS2 protein with the neuron-specific KO. Could it be that neurons have lower SORCS2 transcript levels but higher protein levels than astrocytes? This might explain the loss of impact in cortex in the astrocyte-specific KO. This can be analyzed in more detail.

The reviewer is correct in as much as conditional inactivation of *Sorcs2* in neurons also reduced levels of the receptor transcript in astrocytes (Ext. data fig. 5f). However, the reduction in transcript levels in astrocytes was modest (30%) as compared to neurons (62%) and only seen in female, not male mice. These sex-specific differences likely reflect technicalities in introducing Cre-induced clean neuron-specific *Sorcs2* gene defects in females, prompting us to use male mice for these experiments instead.

As suggested, we now also provide additional immunohistological data, documenting the presence of SORCS2 in neurons and astrocytes, both in hippocampus and cortex (Ext data Fig. 3d-e). Although qualitative, rather than quantitative in nature, these data substantiate the presence of SORCS2 protein in both cell types sorted from mouse brains (Fig. 3b). Unfortunately, the low yield of sorted cells precludes us from directly comparing protein levels in sorted neurons and astrocytes by Western blotting.

Obviously, different receptor levels in various cell types and brain regions may contribute to the differential effects on A β levels seen in hippocampus but not cortex of the asKO

line. As now elaborated in the result section, differences in astrocyte subtype composition or vulnerability, seen in these two distinct brain regions, likely explain the region-specific A β phenotypes in mice with astrocyte-specific SORCS2 loss (page 11, line 12 from below).

While we cannot confidently compare levels of SORCS2 protein levels in astrocytes versus neurons based on transcript levels in sorted cells, our data nevertheless strongly support a decisive role of astrocytic receptor expression in AD. This conclusion is based on receptor transcript levels to specifically increase in astrocytes, but not in neurons, in response to the PDAPP transgene (Fig. 4b). Also A β -induced cell loss is seen in astrocytes but not neuronal cell types in PDAPP/KO mice (Fig. 4c).

8. If the mechanism leading to altered delta secretase relies on astrocytic-expressed SORCS2, why use the full knock-out to study delta secretase? This was not well justified.

Our choice to study δ secretase activity in the full knockout rather than in astrocyte-specific mutant mice was based on the more pronounced phenotypes of late-stage AD observed in the global KO model. These phenotypes include tau and amyloid pathologies, brain inflammation, as well as astrocyte cell loss.

We assume that the differences in the extent of these phenotypes is due to the fact that the global KO strategy permanently disrupts SORCS2 expression in astrocytes during the entire life span of the animals. By contrast, conditional gene inactivation by tamoxifen-induced Cre only temporarily disrupts astrocytic SORCS2 expression from the time of Cre induction (12 weeks of age) to the time of analyses (32 or 38 weeks of age). Following acute Cre-induced *Sorcs2* inactivation, mutant astrocytes are continuously replaced by non-targeted SORCS2 expressing cells, gradually reducing the phenotypic outcome of astrocytic SORCS2 loss.

Our assumption that the conditional KO model represents a model of early-stage AD, is supported by new data included in the revised manuscript. These findings document that early signs of cell stress, such increased levels of GFAP (Fig. 6d-e) and IL33 (Fig. 6g), as well as accompanying cell death (Fig. 6f), are seen in astrocyte-specific PDAPP/asKO mice. By contrast, feature of late-stage AD of the full KO, such as hyperphosphorylation of tau (Ext. data figure 6) or amyloid plaque deposition (not included in the revised manuscript) are not detected. We now more clearly explained our reasoning for the choice of the late-stage AD model in the revised result section (page 12, from top, and line 8 from bottom).

Obviously, phenotypes in the full KO mouse model may also be impacted by loss of SORCS2 expression in neurons. This caveat is now more clearly stated in our manuscript

(page 17, from top). The corresponding text reads: “Our data do not exclude confounding effects from neuronal loss of SORC2 expression on AD pathology in the PDAPP/KO mouse model. However, an increased A β burden, as well as additional phenotypes, such as elevated levels of GFAP and IL-33, and induced cell death, are shared by PDAPP mice with global and astrocyte-specific *Sorc2* gene defect, argue that astrocytic SORCS2 deficiency plays a decisive role in inducing AD pathology.” We trust that these considerations represent a fair and balanced interpretation of our data.

9. How does delta secretase contribute to the phenotypes directly? Is delta cleavage of APP detected in PDAPP mice? Is it possible that the enhanced delta secretase activity is unmasked following a complete loss of SORCS2 that normally does not happen in the context of AD in humans and AD models? This gets back to points 1 and 2 above.

Our original data documented increased δ secretase expression (Fig. 7c-f) and activity (Fig. 7i) in brain tissue extracts of from PDAPP mice lacking SORCS2.

To more directly implicate δ secretase hyperactivity in AD-related phenotypes in this mouse model, we now used mass spectrometry analysis to document processing of APP into sAPP δ in SORCS-deficient brain tissue (Fig. 7g). Furthermore, we used ELISA to also document increased levels of tauN368 in PDAPP/KO mice compared to matched controls (Fig. 7h). TauN368, prone to aggregation, is a cleaved tau fragment produced by δ secretase. Jointly these new data substantiate enhanced δ secretase activity as the possible underlying cause of tau and amyloid co-morbidity in AD models lacking SORCS2.

Concerning the relevance of SORCS2 deficiency for AD pathology in patients, human genetics clearly links *SORCS2* variants with the sporadic form of AD. These findings are now explained in more detail in the introduction section (PMID 23673467). The relevance of SORCS2 expression for human pathology is now further supported by new data in the revised manuscript documenting increased *SORCS2* transcription in brains from AD patients as compared to control subjects (Fig.1j). Obviously, we have no direct proof that these associations operate through aberrant induction of δ secretase activity in carriers of *SORCS2* risk gene variants. Still, a near perfect match of phenotypes in AD patients and SORCS2 KO mice, including amyloid and tau co-morbidity, gliosis, and brain inflammation, unparalleled by any other murine model of AD, strongly argues for the clinical relevance of SORCS2 malfunction.

Minor points

1. Add labeling of age and sex of mice to graphs.

Figure panels have been revised to clearly indicate sex and age of animals used in each experiment.

2. Line 70-71, Figures 2a and 2b, regarding the statement, “PDAPP/KO females showed an age-dependent increase in brain cortex levels of soluble A β 40 and A β 42 compared to PDAPP/WT, starting from 12 weeks of age (Fig. 2a-b).” This statement is a bit misleading in that it suggests that PDAPP/KO mice have significant and consistent changes in both A β 40 and A β 42 from 12 weeks onward. However, PDAPP/KO mice do not show significant changes in A β 40 until 25 weeks. Also, A β 42 is not significantly different at the 16 and 19 week timepoints in PDAPP/KO according to the analysis.

We regret ambiguous description of onset of amyloid burden in these mice. We now have revised the result section to more correctly describe the data in Fig. 2a-b (page 6, line 2 from top), documenting a statistically significant increase in A β levels in mutants at 25 weeks of age.

Reviewer #2 (Remarks to the Author):

The manuscript by Schmidt et al. shows that astrocyte distress induced by A β stress promote A β pathology through δ secretase (AEP). Specifically, they first found that Sorcs2 KO astrocytes are more vulnerable to A β stress both in vitro and in vivo. Therefore, they further use Sorcs2 deletion as a model to study heightened astrocyte stress in AD. SORCS2 deficiency induces A β pathology and tau phosphorylation in AD mice, and astrocytes are identified as the most impacted cell type. The mechanism is explained that Sorcs2 deletion provokes microglia activation and pro-inflammatory response in AD mouse. Therefore, this leads to the activation of δ secretase (AEP) to overproduce A β .

The authors need to address some major unanswered questions and concerns prior to publication:

1. The authors indeed see the increased tau fragments (D421/D422) as tau seeds in PDAPP/KO brain. Tau N368 fragmentation is essential for tau aggregation produced by δ secretase, thereby tau N368 should be detected as major tau seeds in Figure 3 and Figure 6.

We appreciate this helpful suggestion. Accordingly, we now include new ELISA data in the revised manuscript documenting increased levels of cleaved tau_{N368}, as well as tau_{N368}/tau_{total} ratio in PDAPP/KO mice compared to matched controls (Fig. 7h). These findings further implicate increased levels of δ secretase activity as a direct cause of tau pathology in SORCS2-deficient AD brains.

2. Global PDAPP/KO mice show increased tau phosphorylation and fragmentation, however, in the PDAPP/asKO mice, the tau pathology is not shown. As the importance of tau pathology in AD, tau phosphorylation and fragmentation should also be detected for PDAPP/asKO mice.

We now include additional data on tau pathology in PDAPP mice carrying an astrocyte-specific *Sorcs2* gene defect (Ext. data figure 6). In line with a more modest increase in amyloid burden, as compared to the full KO, no significant impact on ptau and tau_{N368} levels were seen by Western blotting or ELISA in the PDAPP/asKO model. However, new data included in the revised manuscript now document additional features of global SORCS2 deficiency being recapitulated in animals with astrocyte-specific loss of SORCS2. Shared phenotypes include increased GFAP levels (Fig. 6d-e) and aggravated cell death (Fig. 6f) in hippocampi. Also, levels of the early stress marker IL33, increased in PDAPP/KO mice at a young age (22 weeks), were also significantly elevated in PDAPP/asKO animals (Fig. 6g).

Jointly, these findings argue that astrocyte-deficient mice represent a model of early signs of A β -induced cell stress, while the obligate KO model exhibits feature of severe amyloid and tau co-morbidity representative of late-stage AD. We suspect that the underlying reason for these distinction in phenotypes is the difference in timing and duration of *Sorcs2* inactivation in both models. As detailed in the revised manuscript (page 12, line 8 from below), the global knockout strategy permanently disrupts SORCS2 expression in astrocytes during the entire life of the animals. By contrast, conditional gene inactivation by tamoxifen-induced Cre only temporarily disrupts astrocytic SORCS2 expression from the time of Cre induction (12 weeks of age) to the time of analyses (32 and 38 weeks of age). Following acute Cre-induced *Sorcs2* inactivation at 12 weeks of age, mutant astrocytes are continuously replaced by non-targeted SORCS2+ cells, gradually reducing the phenotypic outcome of astrocytic loss of SORCS2.

We trust that these revisions provide a detailed explanation for the distinct phenotypes in obligate versus conditional SORCS2-deficient mouse models.

3. A β antibody staining (eg. 6E10) or X34 staining are well-established methods to evaluate A β pathology. At least one of the methods should be used for PDAPP/KO mice in Figure 2 and PDAPP/asKO mice in Figure 6.

As suggested, we now include new data on the plaque burden in mice with global SORCS2 deficiency using 6E10 staining of histological brain sections (Fig. 2i-j). These new data fully corroborate our findings in the original manuscript of a massive age-dependent increase in plaque deposition in PDAPP/KO mice shown by thioflavin S staining (Fig. 2g-h). In line with the more modest increase in soluble A β levels (Fig. 6a-b), as compared to the full KO model, the PDAPP/asKO line does not show a significant increase in plaque burden (data not included in the revised manuscript).

4. In figure 7, antibody specific for APP N585 should be used to detect the cleavage of APP by AEP.

We agree that detection of APPN585 cleavage in SORCS2 KO mouse brains would provide further evidence for enhanced activity of δ secretase in this tissue. As antibodies directed against APP585 were not available to us, we used mass spec analysis to document APPN585 cleavage in SORCS2-deficient brain tissues. Following the protocol described in a seminal publication (PMID 26549211), we incubated recombinant APP with SORCS2KO brain tissue extracts and documented cleavage at N585 by peptide sequencing (Fig. 7g). Taken together with our new data, showing increased levels of tau N368 in SORCS2 KO brains (point 1 above), our revised study provides robust

experimental evidence that increased levels of AEP expression (Fig. 7c-f) and activity (Fig. 7i) results in proteolytic cleavage of its substrates APP and tau.

5. The author claimed the overproduction of A β is caused by activated δ secretase (AEP) in PDAPP/KO mice. Although AEP activation is observed in PDAPP/KO mice, this could only be proved when inhibition of AEP (by genomically deletion or inhibitor) blocks the A β overproduced.

We agree that experiments to block AEP activity in AD models *in vivo* would provide evidence for its causal involvement in AD pathologies in SORCS2-deficient mice. However, applying pharmacological inhibitors to mice will require application for a new animal experimentation license, a procedure that currently takes approximately 12 months until possible approval. Similarly, introducing an AEP gene defect into our PDAPP lines with full or conditional *Sorcs2* defects will require extensive gene targeting and breeding efforts (as well as new animal experimentation licenses), significantly exceeding the timeline set for completion of this study. It is therefore that we wish to refrain from carrying out such extensive new experiments. However, we have more clearly mention the caveat of not having document rescue of SORCS2 KO phenotypes by inactivation of AEP *in vivo* in the revised discussion section (page 19, line 12 from below).

Along these lines, we also wish to direct this reviewer's attention to new data on AEP expression in co-culture models of astrocytes and neurons carried out at the request of reviewer 1 (point 8). These data provide additional strong experimental support for δ secretase as a direct target of SORCS2 action in stressed astrocytes. In detail, using co-culture models of astrocytes and neuroblastoma cells, we now show that loss of SORCS2 in astrocytes, exposed to A β , triggers a stress response in neighboring wildtype neurons to induce expression of δ secretase, the proposed molecular cause of AD pathology in SORCS2-deficient mouse models (Extended Data Fig 8).

Other minor comments:

1. Many figures fails to show which groups the p values are referring to. Besides, the p value for Figure 1F left panel is missing.

We apologize for unclear depiction of statistical data in some figures. We now have revised all display items to clearly indicate the experimental groups that the indicated p values refer to. The p value for data in Fig. 1f is non-significant. We now include this information as n.s. in the figure.

2. *IF images from Figure 1C should be quantificated.*

IF images in Fig. 1C are meant as experimental proof that exogenously applied A β is internalized by wildtype and SORCS2-deficient primary astrocytes. The mere qualitative nature of these immunocytochemical analyses does not allow us to derive solid quantitative data on possible differences in A β uptake between genotypes. Rather, solid quantitative data, documenting increased intracellular accumulation of A β in SORCS2-deficient as compared to wildtype astrocytes, are given in Fig. 1d using ELISA.

3. *Enternal control (eg. GAPDH) should be introduced for Figure 1E, Figure 5E and Figure 6D.*

We now include loading controls for these Western blots in the manuscript.

Reviewer #3 (Remarks to the Author):

The manuscript by Schmidt et al provides an interesting group of studies implicating astrocytic SORCS2 signaling in Alzheimer's disease. Evidence mostly comes from global and astrocyte specific KO of SORCS2 in PD/APP mice. The results show that SORCS2 KO increases GFAP levels, neuroinflammation, and elevated Abeta and phospho-tau levels. The authors suggest that the effects of SORCS2 KO are mediated by the activation of delta secretase.

Many of the SORCS2 KO data are compelling, but there are some major conceptual issues that limit the impact of the results, which are outlined below along with a few technical questions.

1. Evidence showing higher pathology loads in astrocyte-specific SORCS2 KO in mice is compelling, but the actual significance or relevance of this KO to AD is not clear. Does SORCS2 expression change in astrocytes during AD? Human data would greatly strengthen the validity of this pathway in the context of disease. Assessing changes in SORCS2 levels in astrocytes as a function of aging in the PD/APP model would be important to include as well. If SORCS2 changes in astrocytes were important to the pathophysiology of AD, one would expect these changes to occur early in the course of disease.

We fully agree that data on the expression of SORCS2 in the healthy versus the diseased brain will provide valuable additional information on its causal role in AD. Therefore, we now include new data documenting increased expression of SORCS2 in astrocytes, but not neurons, isolated from mouse models of AD as compared to non-AD mice (Fig. 4b). Increased astrocytic receptor expression was already noted in PDAPP mice at 20 weeks of age, prior to full blown pathology (Ext. data figure 4a-b), supporting our hypothesis that induction of SORCS2 expression is an early sign of protective stress response in wildtype astrocytes.

Importantly, we now also document enhanced SORCS2 transcript levels in brain specimens from AD patients compared to non-AD subjects (Fig. 1j). Jointly, these exciting new findings from mouse models and humans provide further support for the proposed role of SORCS2 as protective stress factor, uniquely upregulated in astrocytes in response to amyloid burden.

2. Along the same lines, the title of the manuscript should be changed to reflect that this was a mouse study, and not a study on actual Alzheimer's disease.

As suggested, we revised the title to read: "Astrocytes distress triggers brain pathology through induction of δ secretase in a murine model of Alzheimer's disease"

3. Another conceptual issue is that SORCS2 KO appears to kill astrocytes, but there doesn't seem to be much astrocyte loss in AD (again, this gets at whether the results are relevant to human disease). As astrocytes are protective cells by nature, it seems likely that any manipulation that kills astrocytes would also exacerbate neuropathology?

We now revised the discussion section to cite prior studies that document astrocytes reactivity and astrocyte loss as a feature of AD, findings in line with our data (page 17, line 11 from below). The corresponding text reads: “Prior studies have reported the sensitivity of astrocytes to A β exposure⁴, and a corresponding astrocyte cell death in AD patients' brains^{51, 52, 53}.”

Our study investigated the effect of A β -induced astrocyte distress on brain pathology in a mouse model of AD. Whether unrelated insults to disrupt astrocyte viability will cause comparable phenotype of amyloid and tau co-morbidities, and associated δ secretase overactivity, is a possibility but largely speculative.

4. Technical question: Does SORCS2 KO also kill astrocytes at 20 weeks of age (maybe this explains the opposing results of the KO in 20 week old vs 40 week old mice?)?

We wish to direct the reviewer's attention to data in Fig. 4d in the original manuscript that document comparable numbers of astrocytes sorted from the brains of PDAPP/WT and PDAPP/KO mice at 20 weeks of age. Thus, astrocyte loss seen in PDAPP/KO at 40 weeks of age (Fig. 4c) is a consequence of SORCS2 deficiency that appears with age and progressing plaque pathology.

5. Technical question: The FACs data suggest SORCS2 transcript expression is higher in astrocytes than other neural cell types, but is SORCS2 protein expression enriched in astrocytes? The lack of immunohistochemical/fluorescent evidence is a major weakness of the study.

We now provide additional immunohistological data, documenting the presence of SORCS2 protein in neurons and astrocytes, both in hippocampus and cortex (Ext data Fig. 3d). Although qualitative, rather than quantitative in nature, these data substantiate the significance of *SORCS2* transcripts detected in both cell types sorted from wildtype mouse brains (Fig. 3b). Unfortunately, the low yield of sorted cells precludes us from directly comparing protein levels in sorted neurons and astrocytes by Western blotting.

However, irrespective of whether protein levels in astrocytes are higher or lower than in neurons, novel expression analyses included in the revised version of the manuscript now provide further strong support for a distinctive role of astrocytic *SORCS2* in AD. As shown in Fig. 4b, levels of *Sorcs2* transcript are increased in astrocytes, but not neurons,

sorted from mouse models of AD when compared to non-AD cells. These exciting new data substantiate a proposed role of SORCS2 as stress factor in astrocytes, uniquely upregulated in this cell type in response to amyloid burden.

As for additional immunohistological evidence, we now provide additional data, documenting the presence of SORCS2 in neurons and astrocytes, both in hippocampus and cortex, and the loss of expression in mice carrying a *Sorcs2* gene defect (Ext data Fig. 3d). We also include new immunohistological data refining AD-like pathologies in our SORCS2-deficient mouse model, including the age-dependent extent of plaque burden (Fig. 2g-j) as well as the association of tau aggregates with the brain vasculature (Fig. 3i). These phenotype are shared by AD patients, as stated in the revised text (page 7, line 4 from top), and further corroborate the relevance of our experimental models for the human condition.

6. Technical question: in the FACS sorting protocol, it appears that neurons are identified by process of elimination (negative for CD45, C49a, O4, and ACSA2). Neurons can be tricky to sort from intact brain tissue. Often many of the neuronal processes are lost in these procedures. This could introduce a bias, as many mRNA species are found in neuronal processes. Is it possible that the percentage of neurons expressing Sorcs2 (shown in Figures 4B and C) is underestimated?

We agree with this reviewer's notion that a quantitative comparison of *Sorcs2* transcript levels in sorted astrocytes and neurons may be influenced by sorting protocols. We now revised the text to refrain from statements directly comparing transcript levels in sorted brain cell types in the result section (page 8, line 4 from top). Furthermore, we now clearly state in the discussion that our data document robust expression of SORCS2 in sorted murine neurons and astrocytes, but do not allow us to directly compare transcript levels (page 17, line 7 from top).

Importantly, this caveat does not impact main findings in our study, namely that (i) AD pathology induces *Sorcs2* expression in astrocytes but not neurons (Fig. 4b) and (ii) that SORCS2 deficiency reduces the numbers of astrocytes, but not neurons, in mouse models of AD (Fig. 4c). Thus, regardless of whether transcript (or protein levels) of SORCS2 are higher or lower in astrocytes than neurons, the unique expression regulation in astrocytes in response to A β argues for the central role of astrocytic SORCS2 in AD stress response.

7. Technical issue: the delta secretase results are interesting, but whether these changes occur specifically in reactive astrocytes is highly speculative.

Expression of δ secretase increases with brain age (PMID 26549211) and is seen mainly in neurons in the AD brain (PMID 23640887).

To interrogate mechanisms whereby distress of SORCS2-deficient astrocytes may increase δ secretase levels in neurons, we set up a co-culture model of SH-SY5Y neuroblastoma cells with primary astrocytes from WT or KO mice. In this assay, the co-cultures were treated with media conditioned with A β , and the consequential response in δ secretase expression assayed in astrocytes and SH-SY5Y cells separately. Expression of δ secretase did not increase in WT or KO astrocytes when exposed to A β (Ext. data Fig. 8b-c). However, SH-SY5Y cells responded to A β stress in KO astrocytes with a significant increase in δ secretase levels, an effect not seen when co-cultured with stressed WT astrocytes (Ext. data Fig. 8d). These findings argue for a molecular concept of impaired A β stress response in SORCS2-deficient astrocytes that results in an aberrant increase in δ secretase levels in neurons *in trans*. Intriguingly, this mode of action is in line with established activities of SORCS2 in other biological contexts, whereby loss of receptor expression in astrocytes or pancreatic α -cells causes impaired stress response in endothelial cells in stroke (PMID 31898841) or β -cells under glucose stress (PMID 38226160), respectively. Conceivably, both loss of a protective or gain of a toxic function of stressed KO astrocytes may explain the aberrant induction of δ secretase in neurons.